# Retrieve, Merge, Predict: Augmenting Tables with Data Lakes

**Riccardo Cappuzzo**                                        *riccardo.cappuzzo@inria.fr*
*SODA Team - Inria*
*Saclay*

**Aimee Coelho**                                             *aimee.coelho@dataiku.com*
*Dataiku*
*Paris*

**Felix Lefebvre**                                           *felix.lefebvre@inria.fr*
*SODA Team - Inria*
*Saclay*

**Paolo Papotti**                                            *papotti@eurecom.fr*
*EURECOM*
*Biot*

**Gael Varoquaux**                                           *gael.varoquaux@inria.fr*
*SODA Team - Inria*
*Saclay*

**Reviewed on OpenReview:** *https://openreview.net/forum?id=4uPJN6yfY1*

## Abstract

Machine-learning from a disparate set of tables, a data lake, requires assembling features by merging and aggregating tables. Data discovery can extend autoML to data tables by automating these steps. We present an in-depth analysis of such automated table augmentation for machine learning tasks, analyzing different methods for the three main steps: retrieving joinable tables, merging information, and predicting with the resultant table. We use two data lakes: Open Data US, a well-referenced real data lake, and a novel semi-synthetic dataset, YADL (Yet Another Data Lake), which we developed as a tool for benchmarking this data discovery task. Systematic exploration on both lakes outlines *1)* the importance of accurately retrieving candidate tables to join, *2)* the efficiency of simple merging methods, and *3)* the resilience of tree-based learners to noisy conditions. Our experimental environment is easily reproducible and based on open data, to foster more research on feature engineering, autoML, and learning in data lakes.

## 1 Introduction: learning on data lakes needs feature engineering

New tools keep facilitating data science with machine learning, partly automating it, as with autoML (He et al., 2021; Karmaker et al., 2021; Erickson et al., 2020; Feurer et al., 2022; Hutter et al., 2019). However, they typically take a single table as an input, while data scientists often start from a data lake: a loose corpus of tables (Nargesian et al., 2019; Spotify, 2020; Phan, 2023). Learning then requires assembling multiple tables, which needs either in-depth knowledge of the data, or data discovery (Fan et al., 2023a; Hulsebos et al., 2024). Data assembly is typically studied in scenarios where the schema is known (Robinson et al., 2024; Wang et al., 2024): how columns of various tables are related and which can be merged, as in a relational database. However, data lakes often come without this information. In this work, we consider a data lake

situation where the meaningful merges are unknown and where only a fraction of the tables in the data lake are relevant for the given machine learning (ML) task. Consider the following example scenario:

*Alice, a data scientist, wants to predict the rating given to a movie: she has access to a table with information about movies (e.g., year of release, director, language, budget, …). She wants to complete the table, finding more information on the subject beneficial to her task; for example, joining the table about movies with a table on the cast of those movies might help, as some actors tend to appear in movies with better ratings. Alice has access to some large repository of data, or to some search engine that she can query to find additional tables (Castelo et al., 2021; Google, 2023). However, she does not know the schema describing the relation between tables in the data repository. Her objective is to find the tables that are most relevant to her task, to merge them with the original table and finally to use the improved table to build a model that predicts the movie rating. To ensure reliable prediction results, a cross-validation setup repeats these steps over different train-test splits; as resources (compute, RAM…) are limited and cross-validation has a cost, Alice is interested in finding the optimal set of candidates to achieve the best results within a certain budget.*

Research in this area spans the database and machine learning communities. The corresponding pipeline (Figure 1) uses retrieve, merge, and predict steps that involve four tasks: *1)* **retrieving** tables that are joinable with the original table (Fernandez et al., 2019; Zhu et al., 2019; 2016; Castelo et al., 2021; Fernandez et al., 2018; Dong et al., 2023), *2)* **selecting** which joins should be executed to improve the performance of the subsequent ML model (Esmailoghli et al., 2021; Liu et al., 2022; Galhotra et al., 2023; Deng et al., 2017a), *3)* **aggregating** results in cases of one-to-many or many-to-many joins (Chepurko et al., 2020; Kanter & Veeramachaneni, 2015), *4)* **predicting** with supervised learning on the resulting table, a tabular learning problem (Shwartz-Ziv & Armon, 2022; Borisov et al., 2022; Grinsztajn et al., 2022). Elaborate methods have been designed for each of these tasks separately. Yet, how they contribute to the overall machine learning pipeline is often unclear. Prediction performance is not evaluated consistently in the database literature. Moreover, evaluation on data lakes is particularly challenging. First, there is no openly available reference data suited for this purpose — both a data lake and tables with corresponding analytic tasks are needed. Second, a solid evaluation must run the pipeline many times across loosely structured, messy ensembles of tables. State-of-the-art publications relevant to this pipeline seldom come with implementations robust and scalable enough for cross-validation loops on a data lake. Indeed, data discovery and assembly code can involve many complex elements, such as probabilistic data structures or large language models, which need careful software engineering to be production ready. The difficulty of operating those complex pipelines does beg the question of when the prediction gain is worth the operational cost. State-of-the-art research often

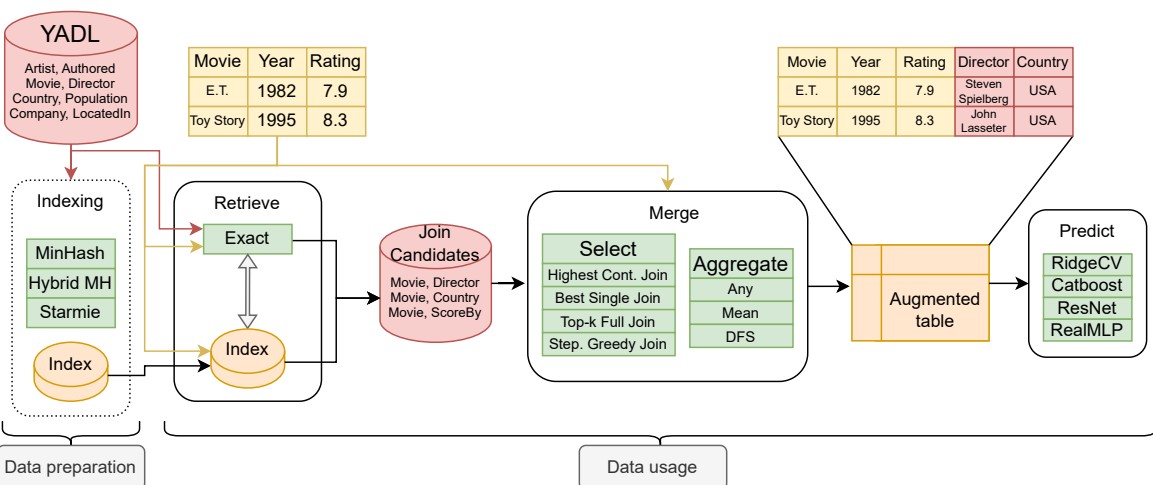

Figure 1: **The evaluation pipeline**. Given a base table, the three main steps (Retrieve, Merge, Predict) augment it with the information from the lake to improve the prediction performance. The data preparation step can be executed offline, and the resulting index may be reused across different data usage instances.

leaves aside the question of complexity, which involves not only computational cost, but also preparation and maintenance burden (Sculley et al., 2015; Paleyes et al., 2022).

We study the full data-discovery for machine learning pipelines, with an eye on exploring the complexity gradient in a reproducible way. For this, we contribute Yet Another Data Lake (YADL), a dataset that enables systematic exploration of the aspects of the pipeline important for good prediction. Built from the YAGO knowledge base (Mahdisoltani et al., 2014), YADL provides a controlled environment to test methods. To analyze the discovery and merge pipelines across different degrees of complexity, YADL's synthesis procedure allows us to vary the shape (number of rows and columns) and degree of redundancy of the resulting tables.

We investigate the impact of the main steps on the end goal of learning from data lakes: preparing the data structures needed to search for candidates, retrieving and selecting potential augmentation candidates, integrating them with the base table, and subsequently evaluating learning on this assembled data. Considering both retrieving and combining tables in a data lake and the downstream learning task, our work bridges database search and machine learning, a topic of much interest (Fey et al.; Robinson et al., 2024; Cvetkov-Iliev et al., 2023; Wang et al., 2024). It has two main objectives: first, **determine which pipeline steps are most important for prediction performance**, and second, develop a level playing field that facilitates **a holistic assessment of learning on data lakes**.

We analyze various methods in the four tasks, with an exhaustive empirical evaluation that required approximately 21 years or 189k CPU and GPU hours (Table 7). Our findings show that *1)* automated selection of joins is noisy, creating very messy tables for which tree-based methods are more robust than deep learning models; *2)* better retrieval of joins reduces the amount of noise and the fraction of null values; *3)* relying on Jaccard containment (i.e., the fraction of entities of the table to augment found in a join candidate) as a criterion for retrieving joins improves downstream prediction performance; *4)* downstream performance plateaus rapidly as more tables are merged, while the resource cost increases; *5)* finally, simpler, metric-based join-discovery methods outpace their complex counterparts in terms of speed and the larger cost of more elaborate methods does not yield proportionate gains in prediction performance.

To validate the generality of our findings, we contrast the insights obtained from the analysis carried out via YADL with those obtained from Open Data US, a well-referenced real data lake (Nargesian et al., 2018; Galhotra et al., 2023): this comparative approach ensures that YADL is representative enough to be a benchmark. We consider 6 input base tables from the literature which pertain to multiple domains.

After outlining the problem setting and previous work in Section 2, we share our main contributions:

1. We develop YADL: a novel benchmarking data lake that allows to test retrieval and augmentation techniques in a controlled environment (Section 3). YADL, the base tables, and the pipeline are available and easily extendable to spur further research.
2. We implement a full prototype pipeline for augmenting tables from data lakes (Section 4). We then measure the prediction performance, execution time, and RAM usage of each method.
3. We conduct an experimental study on 6 base tables over various data lakes, retrieval techniques, join selection methods, aggregation solutions, and ML models (Section 5). Results provide insights on the steps of pipeline learning from data lakes.

We conclude the paper with a discussion of future research directions in Section 6.

## 2 Problem setting and related work

**Problem setting.** Consider a user training a ML model to predict some quantity. The corresponding quantity appears in the training data as a target column $Y$ of a *base table $T$*. Assume our user has also access to a large collection of tables (a *data lake*) $D = \{T_1, T_2, ..., T_m\}$, some of which may contain additional information that can enrich the base table $T$. The data lake may be publicly available (Castelo et al., 2021; Mattmann et al., 2018; Srinivas et al., 2023; Hulsebos et al., 2023; Nargesian et al., 2018) or private (e.g., corporate data). Departing from many studies on learning on relational databases, where schemas are provided (Motl & Schulte, 2015; Fey et al., 2023; Robinson et al., 2024), we consider *unstructured* data lakes,

i.e., data lakes without a schema to specify any PK-FK (Primary Key - Foreign Key) relationship between the tables.

Each of these tables $T_k$ is a bi-dimensional collection of data organized in columns $C_k^i \in T_k$ that may include categorical (names, codes etc.), numerical data (price, revenue, tax rates etc.) and text (product descriptions, user reviews etc.). While cross-table metadata such as foreign keys is not available in this setting, joining $T$ with some of the tables in $D$ would be beneficial for the target prediction task.

Given $T$ and $D$, a table $T_k \in D$ is considered to be *joinable*, or a *join candidate* for $T$ on a query column $Q \in T$, if at least one of the columns $C_k^i \in T_k$ has a non-empty intersection with one of the columns in table $T$: if $\exists Q \in T, \exists C_k^i \in T_k \mid Q \cap C_k^i \neq \emptyset$.

The user wants to optimize the performance of the ML model on a collection of columns (or "features") $X$ according to the quality of the prediction for the target $Y$ (*e.g.,* the movie rating). Columns in $X$ may come either from $T$, or from joined tables in $D$.

Furthermore, we consider a use case where this operation pipeline is wrapped in a cross-validation setup to ensure the reliability of the results: we describe this use case as "Data usage" in Figure 1.

**Different steps.** In such a scenario, the user is likely to rely on the following main steps: **Join Candidate Retrieval**, **Join Candidate Merging** (which includes the **Join Selection** and **Aggregation** tasks), and **Prediction**. Depending on the specific scenario, some of the tasks may be executed in a different order or not at all. In the following, we will drop Join from the names when clear from the context.

Another related problem, which we do not study, is that of finding new *samples* rather than *features* (called "table unionability" in databases): identifying which tables may be "appended" to $T$ to increase the number of rows (Khatiwada et al., 2023a; Fan et al., 2023b; Nargesian et al., 2018).

**Finding the join candidates**  Given a base table $T$ and a data lake $D$, the **Candidate Retrieval** task consists in discovering join candidates (**Retrieve** in Figure 1). This task looks for tables that can be considered as "candidate joins" for the given base table. Different methods have been developed for different scenarios: some "dataset search engines" tackle internet-scale crawls of tables (Castelo et al., 2021; Mattmann et al., 2018); other methods are instantiated on data stored locally (Zhu et al., 2019; Fernandez et al., 2019; Zhu et al., 2016; Fernandez et al., 2018; Khatiwada et al., 2022; 2023b).

While integrating data is not a central focus of the ML literature, augmenting features via joins is recognized as key to ML (Paleyes et al., 2022; Kumar et al., 2016). Still, the blind addition of features may lead to diminishing or negative returns, which led to the development of systems that seek to integrate only correlated features (Esmailoghli et al., 2021; Santos et al., 2021). Alternative systems seek to augment $T$ by synthesizing new features (Kanter & Veeramachaneni, 2015; Cvetkov-Iliev et al., 2023; Zhao & Castro Fernandez, 2022). Recent contributions focus on building graph representations of relational databases to leverage GNNs for learning across tables (Fey et al.; Robinson et al., 2024; Wang et al., 2024). An orthogonal research direction is concerned with the problem of search and augmentation while maintaining privacy (Huang et al., 2023a;b).

Though implementations may vary, retrieval methods used to find join candidates share some similarities: 1) they involve an offline phase to either index the data lake (Zhu et al., 2016; Fernandez et al., 2019), or train a model (Fan et al., 2023b; Dong et al., 2023), and 2) they rely on similarity metrics across columns to select candidate joins. For our experiments, we assume the user has run the offline step prior to the retrieval operation (the "data preparation" section to in our pipeline, Figure 1).

Often, the similarity metric is **Jaccard Containment** (JC), defined as

$$\text{Jaccard Containment} := \frac{|Q \cap C_k^i|}{|Q|}, \tag{1}$$

where $Q \in T$ is a query column in base table $T$, $C_k^i \in D$ is a candidate column in table $T_k$, $|Q \cap C_k^i|$ is the cardinality of the intersection between the two sets, and $|Q|$ is the cardinality of the query set itself. Intuitively, if this ratio is high, then a large fraction of $Q$ is found in $C_k^i$, suggesting that the two columns should be joined. Approximate methods can scale JC-based retrieval to very large databases (Zhu et al., 2016;

Fernandez et al., 2019; 2018). Beyond JC, other metrics have been used, such as top-$k$ set similarity (Zhu et al., 2019), combinations of JC and embeddings-based similarities (Fan et al., 2023b), or other embeddings-based metrics (Cong et al., 2022; Habibi et al., 2020), such as approximate nearest neighbor search (Dong et al., 2023) on model embeddings. Methods based on Jaccard similarity are vulnerable to typos, which led to the development of methods that perform fuzzy or semantic matching such as Fan et al. (2023b); Mundra et al. (2023); Deng et al. (2017b); Abedjan et al. (2015); in this work, we focus on scenarios that do not involve fuzzy matching and thus need to rely on an exact metric to ensure overlap between columns, such as JC.

Retrieval methods are often designed for large data lakes (up to millions of tables Zhu et al., 2019; Fan et al., 2023b) and to maximize recall. However, three issues arise. First, these methods do not assess the relevance of join candidates to the downstream task. While a large containment value may indicate that a join *can* be done, it provides no guarantee that this join is actually useful (Kumar et al., 2016). Second, the number of candidate joins could become too large for practical use. Manually identifying the best candidates is time-consuming, and performing all joins might be too expensive in terms of time/memory constraints (Santos et al., 2021). A user-defined threshold on the containment can filter out the least promising joins, but deciding the correct threshold is problematic; alternatively, it is possible to select only the top-$k$ candidates by containment. Third, Jaccard containment does not take into account the cardinality of a column: in an extreme case, if a column $Q \in T$ contains only a single value, it would have perfect overlap with any column $C_k^j \in D$ that contains the same value. While this behavior may be desirable for some specific retrieval tasks, it further expands the number of candidates with high containment. The presence of duplicate tables worsens each of these issues by introducing potential false positives.

To make retrieval readily useful, the **Join Selection** task *identifies a subset of candidate joins that maximizes the prediction performance over a downstream task* (**Select** in Figure 1). These methods use various strategies to add value to retrieval: profiling candidate joins according to various metrics (Galhotra et al., 2023; Flores et al., 2021), rules to remove joins that are not useful (Kumar et al., 2016; Shah et al., 2017), joining over sketches or coresets of the data (Wang et al., 2022; Santos et al., 2021), or executing each candidate join to find those that bring benefit (Esmailoghli et al., 2021; Chepurko et al., 2020; Liu et al., 2022; Galhotra et al., 2023; Gong et al., 2023; Dong & Oyamada, 2022). In this work, we focus on the latter, very popular, strategy. We consider ML models that operate on rows as samples, columns forming features. Augmentation enriches features while keeping the original set of samples, thus requiring a *left join*. A challenging consequence is that un-joined rows in the left table will lead to missing (null) values in the new columns.

We split Candidate Join Retrieval and Selection to tease out the retrieval metric from the benefit for the task: the first measures similarities between columns, while the second depends on the information in the foreign table.

**Merging the candidates** After the set of join candidates has been retrieved, they must be merged with the base table to augment it with their columns. It is not always possible to limit joins to one-to-one relationships; often, we join on one-to-many or many-to-many relations. For example, to join a table about movies with one that contains movie ratings on column "movie title", every movie with more than one rating is in a one-to-many relation, thus the content of all the rows in such relations gets duplicated[1]. Sample duplication is problematic when the downstream task involves a ML or statistical analysis method, as each row corresponds to one sample.

The **Aggregation** task bridges the result of the join with the downstream methods, combining the information contained from a potentially large number of rows into one (**Aggregate** in Figure 1). More precisely, given a tuple (row) $t$ in the base table $T$ that needs to be joined with $n$ tuples from a table $(A, B)$ over $A$, the goal is to augment $t$ with attribute $B$ by selecting one value that represents the information from the $n$ joining tuples. How to select a representative value for the new attribute reminds data integration problems such as truth discovery (Dong et al., 2009) or data fusion (Bleiholder & Naumann, 2009). In this spirit, Deep Feature Synthesis (DFS) (Kanter & Veeramachaneni, 2015) takes a set of tables and a join plan to recursively aggregate replicated instances using functions such as average, median, and mode.

---

[1]More detail is provided in Appendix subsection B.3

Depending on the join selection strategy, aggregation may be executed during the selection process, or after the set of candidates has been chosen. In any case, it is paired with the selection procedure by the need of training the ML model on the augmented tables. To represent this coupling, we combine the two tasks in a single **Merge** step in Figure 1.

**Learning on the augmented data**   Finally, the integrated table is used to train a model to **Predict** a target variable through supervised methods. Tabular machine learning is the subject of very active research, with well-established tree-based methods (Grinsztajn et al., 2022; Prokhorenkova et al., 2018) and rapidly progressing tabular deep learning (Holzmüller et al., 2024; Ye et al., 2024; McElfresh et al., 2023; Hollmann et al., 2022; Hegselmann et al., 2023; Arik & Pfister, 2021; Gorishniy et al., 2024). It is not possible in our study to explore all recent methods, in particular given that some incur compute costs incompatible with our extensive evaluation and the combinatorics of pipelines. We consider three broad range of methods: *1)* linear regression/classification (as implemented in scikit-learn Pedregosa et al., 2011), *2)* tree-based methods such as CatBoost (Prokhorenkova et al., 2018), *3)* or deep learning methods well suited for tabular data, such as ResNet (Gorishniy et al., 2021) or RealMLP (Holzmüller et al., 2024). While we present the prediction task as distinct from the prior ones, some pipelines implement learning at intermediate sections of the workflow to ensure the selection of the most appropriate candidate tables (Galhotra et al., 2023; Huang et al., 2023c).

## 3  Building Yet Another Data Lake

Yet Another Data Lake (YADL) is a semi-synthetic data lake built by recombining the data present in the YAGO (Suchanek et al., 2007; Mahdisoltani et al., 2014) knowledge base (KB) [2] to generate a collection of tables. Our goal is to have a scalable, high-quality and consistent data lake that allows users to evaluate the main steps of the pipeline in Figure 1, while avoiding some of the confounding factors that come from working with unvetted data, such as typos, inconsistent schemas and data format, and other sources of noise. These challenges can be added to YADL, e.g., by generating typos in table entries.

**YAGO: the source of the original data.** YAGO (Suchanek et al., 2007) is a KB composed of RDF triplets; each triplet has a *subject* connected to an *object* through a *predicate* (or relation). Subjects and objects are considered to be *entities*, e.g., in the triplet "Paris, locatedIn, France", "Paris" is the subject entity, "France" is the object entity, and "locatedIn" is the predicate that connects them. The object may also be a lexical value such as the one for the "population density". Each entity belongs to a set of *classes* (or types), arranged in a taxonomy; "Paris" belongs to the class "City", subclass of "Populated place", itself subclass of "Geographical locations".

**From knowledge base to relational tables.** Information in data lakes is typically stored as tables, rather than the triplet format of YAGO. For this reason, we first rearrange the YAGO triplets by converting them into binary tables, then we filter and recombine the binary tables to create multiple YADL variants (*Base*, *10k* and *50k*) that differ in their size and in the properties of the tables contained therein. YADL variants can be obtained by changing the parameters used for the synthesis, as detailed in Appendix A.

## 4  Implementations of the retrieve-merge-predict pipeline

We now discuss the pipeline steps in Figure 1. For the **Retrieve** step, we discuss different possible retrieval strategies; for the **Merge** step, we propose different join selectors and aggregation methods; finally, for the **Predict** step, we go over the ML methods used to test the prediction performance. We explore each section in more detail in Appendix B.

**Retrieving the candidates.** We consider retrieval methods that work by taking a query column and return a – possibly ranked – list of candidates.

We consider four retrieval strategies that explore different approaches and trade-offs. We first distinguish between"offline" and "online" methods: methods in the first category require some degree of offline data

---

[2]We use YAGO 3.0.3 (Mahdisoltani et al., 2014), which is updated to 2022.

preparation in order to be executed, such as constructing data structures, or training a specific model; methods in the latter category are instead executed directly "online" as part of the "Data usage" section.

**Exact Matching** is on "online" method, and measures the *exact* Jaccard containment between it and every column in the data lake. The "offline" category includes **MinHashLSHEnsemble** (MinHash) (Zhu et al., 2016), **Hybrid MinHash** (a method we develop), and **Starmie** (Fan et al., 2023b). MinHash builds an index on the data lake, and when queried returns candidates whose Jaccard containment is greater than a certain threshold *without ranking them*. Hybrid MinHash combines Exact Matching and MinHash by taking the candidates returned by MinHashing and measuring their exact containment, thus limiting the number of columns to be considered and reducing the computational cost of Exact Matching, whilst introducing a candidate ranking. Finally, Starmie builds embeddings for the query column and every column in the data lake using a language model, then combines cosine and Jaccard similarity to rank candidates. All methods share the same drawback of requiring some degree of updating/retraining as a data lake evolves over time More detail is provided in Appendix B.1.

The "Data usage" part of the pipeline (1) is transparent to the retrieval method: given a list of join candidates, it will test each candidate regardless of how the list was constructed. To reflect practical limitations on runtime and compute resources, we introduce a *budget constraint* on the number of retrieved candidates to evaluate. We choose to keep the top-30 candidates for most of our experiments, as we observe that almost all candidates with high containment are within this limit (Figure 7 in Appendix), it has empirical confirmation in practice (Spotify, 2020), and our experiments show that results plateau past this threshold (Figure 4). In Section 5, we explore in detail how the value of $k$ affects scalability and results.

**Merging base table and join candidates.** In this step, the retrieved candidates are combined into a new *integrated* table that joins the information in the base table $T$ with additional information from augmentation tables. As shown in Figure 1, the merge step combines **join selection** and **aggregation**, as both must be executed to build the augmented table used for Prediction.

**Join selection** We consider two different overarching strategies for selecting candidates: **metric-based selectors**, which rely on a metric to choose which candidates should be joined, and **results-based selectors**, which instead iterate over candidates and select the best by explicitly executing joins. The first category includes **Highest Containment**, which ranks candidates by containment, then joins the first, and **Full Join**, which relies entirely on the ranking provided by the retrieval method to join the candidates without filtering them. For the second category, we implement **Best Single Join** and **Stepwise Greedy Join**. The first works with one candidate at each iteration, during which it first joins the candidate, then it trains and evaluates a prediction model on the resulting table; after iterating over all candidates, it selects the single candidate with the best performance. The second method iterates over each candidate like in the previous case, however it retains all candidates that improve the prediction performance, so that the augmented table grows over time as new candidates are joined. Stepwise Greedy Join is related to forward feature selection (Guyon & Elisseeff, 2003), as it greedily adds new tables as features during each iteration, and represents a common approach in the database literature (Galhotra et al., 2023; Chepurko et al., 2020). Appendix B.2 details the implementations.

**Aggregation** We test three aggregation strategies: **Any** selects any row at random from each group of matched tuples; **Mean** replaces all duplicated numerical (categorical) values by the mean (most frequent) of all values for that attribute in the group; finally **DFS** (Deep Feature Synthesis (Kanter & Veeramachaneni, 2015)) greedily generates new features for each column in the augmented table by aggregating groups of tuples, measuring statistics over the groups (e.g., mean, median, most frequent value), and adding said statistics as features; this is done for every join. We expand on this in Appendix B.3.

**Supervised learning with a ML model.** The learning step is performed on the final, *integrated table* obtained by merging the candidates on the training split of the base table. We evaluate four predictors. We use a **Ridge regressor/classifier** as a linear baseline using the scikit-learn RidgeCV (Pedregosa et al., 2011) implementation and default parameters. **CatBoost** (Prokhorenkova et al., 2018) is a state-of-the-art GBDT method, optimized to handle categorical variables. **ResNet** (Gorishniy et al., 2021) is our baseline neural method. **RealMLP** (Holzmüller et al., 2024) is a NN-based method that incorporates a number of "good

defaults" to improve performance over standard NNs. For ResNet and RealMLP we use the implementation and default parameters provided by pytabkit (Holzmüller et al., 2024). We do not perform hyperparameter optimization. Further detail on implementation details, pre-processing of the data, and cross-validation setup are provided in Appendix C.2.

## 5 Experimental study

### 5.1 Settings

For our experimental campaign, we test the different sections of the pipeline over three dimensions: prediction performance, execution time, and RAM usage. Focusing on multiple dimensions allows us to have a view of the different trade-offs across methods.

**Data lakes**  We use four YADL variants (Binary, Base, 10k, 50k) (as detailed in Section 3) and Open Data US, a data lake employed in the literature (Galhotra et al., 2023; Zhu et al., 2019; 2016); statistics are reported in Table 1. Starmie did not run on YADL 50k (not enough RAM) and Open Data (noise in the data crashed the model), so these data lakes are excluded from figures that involve Starmie. Appendix Figure 7 reports the measured containment for the base tables used in our experiments.[3]

Table 1: **Statistics for the tables in the data lakes**. "N. cols" and "C. cols" refer to numerical and categorical columns, respectively.

|  | YADL Binary | YADL Base | YADL 10k | YADL 50k | OpenData US |
|---|---|---|---|---|---|
| N. tables | 70 | 30072 | 10059 | 47223 | 5591 |
| Size (MB) | 629 | 10051 | 5557 | 27718 | 4062 |
| Tot. rows | 20.1M | 672M | 242M | 1.20B | 95.7M |
| Tot. cols | 140 | 95.2k | 127k | 624k | 133k |
| Avg rows | 287k | 22.3k | 24.0k | 25.4k | 17.1k |
| Avg cols | 2.00 | 3.17 | 12.63 | 13.21 | 23.86 |
| Avg N. cols | 0.30 | 0.39 | 3.54 | 3.59 | 11.10 |
| Avg C. cols | 1.70 | 2.78 | 9.08 | 9.61 | 12.76 |
| Avg nulls | 0.00 | 0.31 | 0.62 | 0.64 | 0.09 |

**Base tables**  We evaluate four tables from sources external to the lakes: Company Employees, US Elections, 2021 US Road Accidents, and Housing Prices. As "internal" tables, derived from lakes, have been used in several previous works for evaluation (Galhotra et al., 2023; Chepurko et al., 2020), we also include one per lake: US County Population (from YADL) and Schools (from Open Data US). Statistics are reported in Appendix Figure C.7 .

For all datasets, the values of the query columns must be matched with the entities in YADL using semantic annotation solutions (Huynh et al., 2022; Nguyen et al., 2021). In our experiments, we manually performed the match to remove the noise from this task. Also, query columns are chosen based on what a user may reasonably consider as "key" (e.g., the company name). We release the matched tables along with the data lakes for reproduciblity.

To reflect the experience of a data scientist that needs to construct a meaningful table starting from a data lake, and to highlight the effect of joins on the downstream task, we run experiments on a depleted version of the tables, i.e., the input tables include only the primary key column and the target column. We provide details on the datasets, computational resources, cross-validation setup, and pre-processing in Appendix C.2.

### 5.2 Results

Our first goal is to pinpoint which steps of the pipeline have the most significant impact on the studied dimensions: optimizing these influential sections can yield the greater benefits. We rely on Pareto optimality plots (Figure 2) to highlight the cost-performance trade-off for the different steps and configurations.

**Reference configuration**  Based on Figure 2, we define a **reference configuration** that uses Exact Matching, Best Single Join, aggregation Any, and CatBoost. This configuration represents a good trade-off between performance and compute cost, and it can run on all configurations, base tables and data lakes.

---

[3]The data lakes are available at `https://doi.org/10.5281/zenodo.10600047`, the code to prepare YADL is at `https://github.com/soda-inria/YADL` and the pipeline is at `https://github.com/soda-inria/retrieve-merge-predict`.

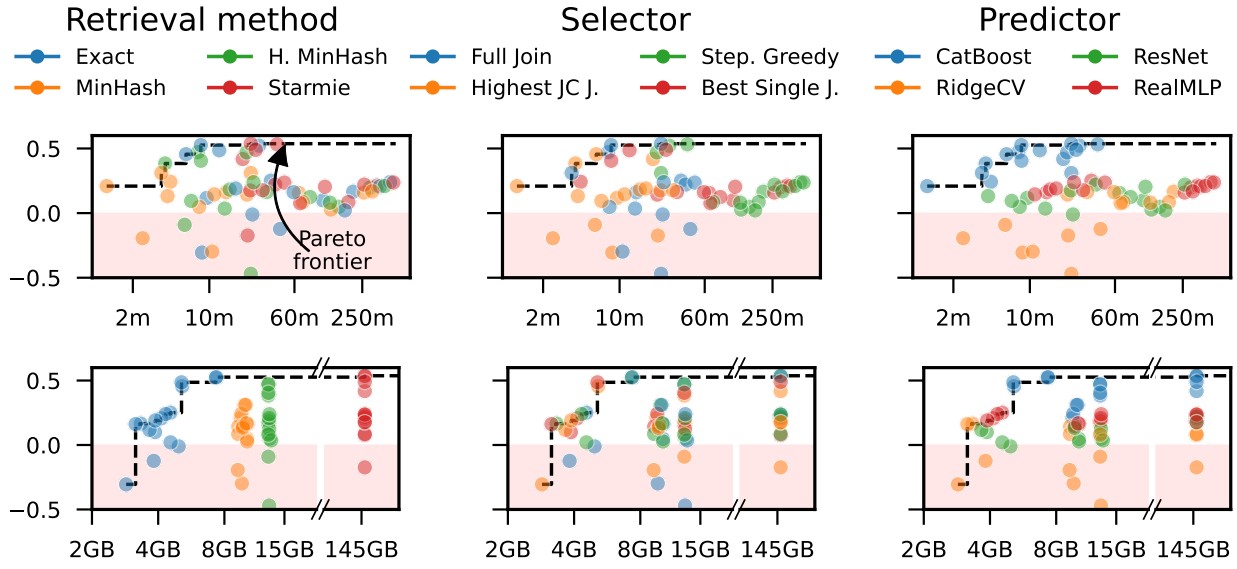

Figure 2: **Pareto diagram for the pipeline steps** The prediction performance ($y$-axis) is plotted against retrieval + run time (top) and peak RAM usage (bottom). Each row presents the same results, broken down by retrieval method (left), join selector method (center) and predictor (right). Each dot represents the average prediction performance and resource cost averaged across base tables and data lakes for a specific configuration (e.g., the leftmost dot in the first row is obtaining by using MinHash, Highest Containment Join and CatBoost). Time for offline retrieval preparation for MinHash and Starmie is not reported here.

Table 2: **Ablation study, median of the prediction difference (higher is better) and runtime difference (lower is better) from the reference configuration. Retrieval Method** does not include results from Open Data US and YADL 50k.

| Retrieval | Diff. Prediction | Diff. Time | Selector | Diff. Prediction | Diff. Time |
|---|---|---|---|---|---|
| Exact Matching | 0.00 % | 1.00x | Full Join | 2.09 % | 0.69x |
| Starmie | -0.06 % | 2.20x | Stepwise Greedy Join | 1.61 % | 2.10x |
| Hybrid MinHash | -3.19 % | 0.67x | Best Single Join | 0.00 % | 1.00x |
| MinHash | -17.48 % | 0.34x | Highest Containment Join | -0.10 % | 0.45x |
| Aggregation | Diff. Prediction | Diff. Time | Predictor | Diff. Prediction | Diff. Time |
| DFS | 2.46 % | 4.42x | CatBoost | 0.00 % | 1.00x |
| Mean | 0.02 % | 1.02x | RealMLP | -21.24 % | 17.52x |
| Any | 0.00 % | 1.00x | ResNet | -24.11 % | 5.92x |
| | | | RidgeCV | -26.03 % | 3.31x |

Table 2 reports an ablation study comparing the performance and runtime of the reference configuration against the other possible methods; we report the median difference in prediction and runtime. [4]

**Predictor: Tree-based methods perform well**  Supervised learning with the ML model is the pipeline step with the starkest difference between methods (Figure 2 right, Table 2): in our scenario, CatBoost is both *faster* and *more effective* than its non-tree counterparts. Indeed, CatBoost always outperforms other methods in prediction performance (by up to 26%) and run-time (17× faster than the slowest competitor).

---

[4]This is expanded in Appendix subsection C.2 and Figure 14.

This is likely due to two major factors: 1) imperfect joins introduce a large fraction of missing values (database "nulls"), and 2) categorical features have high cardinality. Firstly, any sample that does not find a match in a candidate will have a missing value in each new feature; in addition, any missing value *in matched rows* will remain: as a result, even "good" joins may feature a high degree of missingness. Parametric models are particularly affected by missing values, and categorical features make imputation more difficult; comparatively, tree-based models are far more resilient in presence of missing values (Josse et al., 2024). In fact, we observe that CatBoost and RealMLP are the only methods whose prediction performance never drops below 0 even in this very challenging scenario. McElfresh et al. (2023) also report that heavy missingness impedes neural networks more than tree-based models.

**Retrieval: Containment is key, though imperfect**
A surprising result of our experiments shown in Figure 2 and Table 2 is that simple metric-based retrieval is sufficient to achieve good results, while maintaining a far lower resource cost. Relying directly (or indirectly) on Jaccard containment (Highest Containment, Full Join, Exact Matching) achieves the same results as far more thorough (and costly) retrieval (Starmie) or selection (Best Single Join, Stepwise Greedy Join).

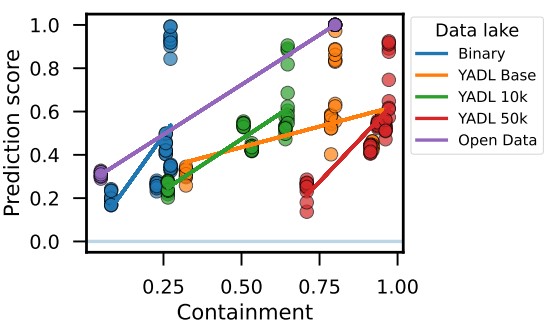

Focusing on the reference configuration, Figure 3 relates containment to prediction performance. The plotted lines highlight the clear upwards trend in the prediction performance that follows the increase in containment. In other words, higher prediction results occur more frequently when the containment is higher. For exact joins (as in our scenario), having a high Jaccard containment leads to better results because left joins with high overlap augment

Figure 3: **Retrieval: Better containment improves prediction performance**. Regression plot relating the prediction performance with the Jaccard containment; each dot represents an experimental run for each base table.

a large fraction of the base table, whereas low overlap joins produce features that contain mostly empty values.

And yet, while Jaccard containment is an effective first metric, it is not fool proof. We observe experimentally that it does not work well where there is a large degree of redundancy because in such cases many candidates may share the same containment: this is what happens with YADL 50k (Table 3, Appendix Figure 7); a larger computational budget may mitigate this issue, but it is not always an option. In other words, *high containment means that we can join without adding too many missing values: this does not guarantee that the added features improve downstream utility.*

**Retrieval: Better metrics improve performance, at a cost**  Pareto diagram (Figure 2, left) shows that Starmie is Pareto optimal for some (resource-intensive) configurations: if resources are available for the offline preparation cost, better querying through Starmie can bring benefits. And yet, the –arguably small– improvements in downstream performance brought by Starmie come with a very large cost in both RAM and offline training time (Figure 2, Figure 22, Appendix Table 6, Appendix Table 10): Starmie has about 7.5× the RAM footprint of the most expensive alternatives; furthermore, it could only run on 3 out of our 5 data lakes of interest (Binary, YADL Base and YADL 10k).  On the other hand, Exact matching is Pareto optimal in most cases, and runs on all data lakes we considered.

With respect to MinHash, Hybrid MinHash brings a substantial improvement in terms of downstream performance (Figure 2, Table 2, Figure 14). This comes at the cost of an increase in the query time (Figure 18, Appendix 10), which is however amortized due to the cost of performing cross-validation.

The cost of retrieving candidates by testing many query columns is another important factor in the scalability of the retrieval methods. Indeed, if the user does not know which is the best query column to use, or if there are multiple, they may favor methods that have a shorter query time (such as MinHash and Hybrid MinHash) in order to run the evaluation pipeline faster. We explore scalability in more detail in Appendix C.1.

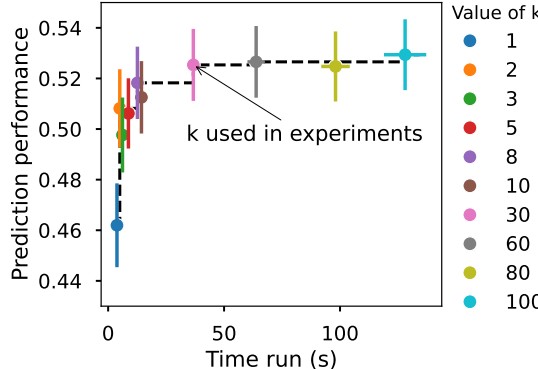

Figure 4: **Trade-off in prediction performance as the number of candidates increases** for the Full Join selector. Results are averaged over all base tables and data lakes and represented as a Pareto plot.

**Retrieval: performance plateaus quickly as $k$ increases**  To understand the impact of $k$ on the prediction performance, we record the prediction performance for a range of values, and report the results in Figure 4. While there is some variability due to the data lake and base table, we observe that results plateau quickly as $k$ increases, saturating for $k$ as low as 8. We explore this facet in more detail in Appendix C.10.

**Aggregation: Complex aggregation improves performance with limitations**  Figure 5 reports aggregation results[5]. The systematic expansion of multiple aggregations in DFS leads to many features, which would make both Full Join and Stepwise Greedy Join intractable, so they are not considered here. The generated features overall bring useful information, as DFS outperforms the simpler aggregations in most cases; however, preparing and employing the new features increases noticeably the training time. This is confirmed by Table 2, which highlights both the improvement in performance ( 2.5%) and the major increase of compute cost ( 4.42x wrt Any).

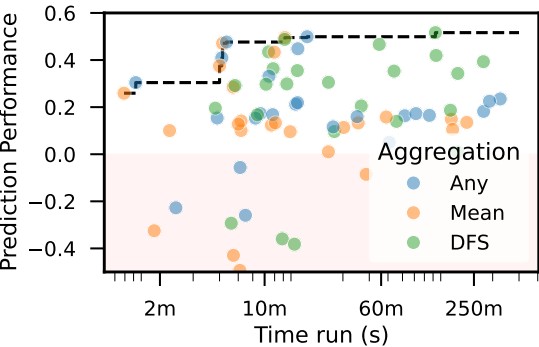

Figure 5: **Aggregation: Pareto diagram.** DFS outperforms other methods, but is much slower.

**Overall results across tables**  Table 3 reports the median prediction performance across all configurations for a pair "Base table - Data Lake". From this table it is evident that internal tables (Schools, US County Population) represent a much simpler problem to solve compared to tables that are not sampled from the data lake (all other tables).

---

[5]More detail is available in Appendix, Figure 15

Table 3: **Aggregated median prediction results (higher is better) for the reference configuration.** "NA" results are referring to experiments run on "internal" tables ("Schools" was sampled from Open Data, "US County Population" from YADL): if run on a different data lake, no matching entities could be found, so no new features would be added.

| Base Table | Binary | YADL Base | YADL 10k | YADL 50k | Open Data US |
|---|---|---|---|---|---|
| Company Employees | 0.20 | 0.33 | **0.37** | 0.25 | 0.26 |
| Housing Prices | 0.34 | **0.57** | 0.54 | 0.54 | 0.50 |
| Schools | NA | NA | NA | NA | **1.00** |
| 2021 US Accidents | 0.26 | **0.44** | **0.44** | 0.43 | 0.31 |
| US County Population | 0.93 | 0.84 | **0.95** | 0.85 | NA |
| US Elections | 0.44 | 0.55 | 0.52 | 0.52 | **0.59** |

# 6 Discussion and conclusions

**Take-away messages**  We build a semi-synthetic data lake based on a knowledge base to use as a reproducible test bed for evaluating methods to augment user-provided tables. We implement an easily-extendable pipeline to test the different steps of the augmentation procedure and alternative algorithms for each of their tasks. Our results uncover a number of observations important to direct research on this subject. We summarize the main take-away messages from the experiments.

1. **Tree-based models** come with significant benefits in terms of prediction and computational performance in the data-lake pipeline we studied (Figure 2 right, Table 2). Indeed, the automated selection of augmentation candidate tables generates challenging features (*e.g.,* with many missing values), which tree-based models are more effective at dealing with.
2. **Good table retrieval** affects the entire pipeline by discovering candidates that contain more useful features and introduce fewer missing values (Figure 2 left). Jaccard containment is a good metric for retrieving candidates, but it has limitations (Figure 3).
3. **Simple metric-based retrieval** (Exact matching) and candidate selection (Highest containment) produce comparable or better results than more complex methods (Starmie[6] and Best single join), while being vastly more efficient (Figure 2 left, center, Table 2).
4. **Performance plateaus quickly** as the number of retrieved candidates increases, while the resource cost increases steadily (Figure 4).
5. Complex aggregation methods are much slower than simpler ones and do not result in commensurate gains in prediction performance (Figure 5, Table 2).
6. Combining stochastic retrieval with exact metrics (Hybrid MinHash) mitigates some of the drawbacks of basic MinHash and scales well with many query columns (Figure 3, Table 2).

The topic that we have studied is at the intersection of database and machine learning research. This intersection still has many open research prospect:

**Merging on clean columns is only one side of the story**. While we focus on columns where join keys can be matched exactly (functionally, by doing string matching), this is not possible in general because of typos, different formats, and different granularity. Similarity joins and semantic matching would help with this problem (Jiang et al., 2014; Dong et al., 2023; Deng et al., 2017b; Mundra et al., 2023) and add another dimension to the analysis. Methods that cover multiple steps, such as (Gong et al., 2023; Huang et al., 2023c), could also be considered.

**To limit the search space, we keep our chain of joins for augmentation limited to one**, and show that scalability is an issue even in this simplified scenario. However, some methods have considered chains of join for augmentation (Galhotra et al., 2023). Enabling join chains would also make evident the benefit of recursive methods (Cvetkov-Iliev et al., 2023; Kanter & Veeramachaneni, 2015).

**"AutoRetrieval" is an exciting prospect.** The overall middling-to-poor prediction results (Table 3, Table 8) suggest that the automated strategies evaluated here are not sufficient to replicate the performance achieved by a human data analyst. Automated procedures may help with discovering good candidates, but they cannot replace a human expert: fully automating this operation similarly to how AutoML tools (Hutter et al., 2019; Feurer et al., 2015) tweak ML methods represents a compelling direction for future work.

**Final words.** The performance-complexity trade-offs that appear are crucial for the practitioner: they suggest how to automate supervised learning on a data lake of a given size, maximizing statistical performance within a compute budget.

**Acknowledgements**  We acknowledge the invaluable insight and expertise provided by Léo Dreyfus-Schmidt and Du Phan (formerly from Dataiku), without whom the foundations of this work would not have existed.

---

[6]Note that our results are not in contradiction with the Starmie study, as it was focused on *table unionability*, i.e. finding more samples rather than feature augmentation which is our focus.

As YADL is based on YAGO 3, we acknowledge the work made by its authors to prepare the original knowledge base, as well as their efforts manually evaluating it. This research was supported by the Project P16, ANR-23-RDIA-0001. We express our gratitude for the financial support that made this study possible. Finally, we would like to thank the anonymous reviewers for their constructive comments and valuable suggestions, which have significantly improved the quality of this manuscript.

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

Table 4: "City" seed table built by flattening predicates.

| Subject | locatedIn_1 | locatedIn_2 | owns_1 | density |
|---------|-------------|-------------|--------|---------|
| Paris | France | Europe | Tour Eiffel | null |
| Rome | Italy | null | Olympic Velodrome | 2236 |

Table 5: "City" seed table built by left-joining binary tables.

| Subject | LocatedIn | Owns | Density |
|---------|-----------|------|---------|
| Paris | France | Tour Eiffel | null |
| Paris | Europe | Tour Eiffel | null |
| Rome | Italy | Olympic Velodrome | 2236 |

## A  Construction of YADL

We construct YADL by first reshaping the RDF triplets found in YAGO into binary tables, then, depending on the construction strategy, we assemble new YADL variants by joining binary tables on each other.

**Binary tables**  A binary table is generated for every predicate in YAGO, e.g., "hasCapital". YAGO contains 70 predicates, which result in a small data lake where some of the tables have millions of rows (e.g., "isLocatedIn") and some have few (e.g., "hasTLD"). Each table contains an attribute named "subject" and another attribute named as the actual predicate. For example, the triplet "France - hasCapital - Paris" leads to a table "hasCapital" with columns "subject" and "hasCapital"; finally, a row "France, Paris" is added to the table. The same process is applied to all 70 relations and their triplets.

**Wordnet-based tables**  To reflect that tables typically have more than two columns, we develop a second variant of YADL where tables have a larger number of columns. To this goal, we leverage the Wordnet (Miller, 1994) classes (e.g., "Person", "Company", "Artist") to which entities belong. We use all Wordnet classes (a total of 1015) to create our *seed tables*, following the example in Table 4. Initially, the seed table includes solely the "Subject" column (e.g., the "City" table includes only values "Paris," "Rome," etc.). Then, we join every relation associated with subjects in the table: in Table 4, the subjects with type "City" are joined with relations "locatedIn", "owns", "hasPopDensity".

Transforming triplet tables into wide format tables introduces null values because not all entities of a given type have the same relations. For example, column "hasPopDensity " contains null values for subject "Paris" in Table 4. The fraction of missing values in the transformed tables is reported in Table 1.

**One-to-many relations**  A subject may be linked to many objects through the same predicate, e.g., "Paris" is linked to two objects ("France", "Europe") via the predicate "locatedIn". We address the issue in three ways. For Binary, triplets that share the same subject and predicate are transformed into tuples with the same value in the "subject" column and different values in the predicate column, e.g., the binary table has columns "Subject" and "isLocatedIn" and it contains tuples (Paris, France) and (Paris, Europe). For "Wordnet" seed tables, we build two variants. In the first case (which we dub *YADL Base* in Section 5), we flatten the group of objects by creating new columns, thus moving them on the same tuple rather than splitting them (e.g., in Table 4, column "locatedIn" is flattened over "locatedIn_1" and "locatedIn_2"). In the second variant (*YADL 10k* and *YADL 50k*), we create the new columns by executing a left join between the subject and each binary table. One-to-many matches (such as with "Paris" and "France", and "Paris" and "Europe") are replicated with each left join; ordinarily, this would lead to an explosion in the number of rows: we limit the issue by selecting only the first two entries for each predicate (as shown in Table 5).

As a result of handling these relations differently in each version of YADL, the aggregation step in the pipeline affects them in different ways, thus exposing different problems.

**Building more sub-tables**  We create supplementary tables derived from the Wordnet seed tables to augment the table count in the data lake with two different methods. In real-life lakes, numerous versions or variants of the same table are common, encompassing collections of tables that undergo slight modifications over time (Halevy et al., 2016). This redundancy poses a challenge for information retrieval methods in distinguishing between pertinent and extraneous tables. Hence, it constitutes a crucial aspect in a benchmark data lake like YADL, which incorporates it heavily in the generation of sub-tables.

For YADL Base, we generate all combinations of arity 2 and 3 for each seed table. For Table 4, these combinations would include "(isLocatedIn_1, owns_1)", "(owns_1, popDensity)", and "(isLocatedIn_1, owns_1, popDensity)". Each sub-table is then built by projecting the seed table over the generated combination; we retain rows that contain at least one non-null value and drop sub-tables with fewer than 100 rows. For YADL 10k and 50k, we drop all seed tables whose arity is lower than a minimum arity $A$; then, for each surviving seed table $T$, we generate $N$ sub-tables by sampling a random subset of columns of size $[A-2, A]$ and projecting onto $T$; each parameter can be tweaked to modify the size and number of resulting sub-tables. Row-wise redundancy is provided by randomly sub-sampling a fraction $p$ of each resulting sub-table $n_s$ more times, thus replicating a fraction of the samples while keeping the set of columns fixed. We construct YADL 10k and YADL 50k by setting $N$ to 10 and 50 respectively, $A = 8$, $p = 0.7$, and $n_s = 2$.

# B  Detailed description of the tested methods

## B.1  Candidate retrieval

**Exact Matching**  We compute *Exact Matching* (Exact) by measuring the exact Jaccard containment for each pair (query column, candidate column) in the data lake. This can be implemented efficiently by first building a "vocabulary" on the query column, and then scanning the whole data lake to compute containment. Candidates are ranked by highest containment, then we select the "top-$k$" candidates in the pool. An advantage of this method is that computing the containment is an embarrassingly parallel operation.

The main drawback of Exact Matching is the computation, whose cost depends directly on the size of the data lake and the tables therein; furthermore, the operation must be repeated for every new query column. As we show in Figure 18, the cost of querying multiple columns quickly adds up. Another drawback is that containment is less reliable if the data lake features a lot of redundancy (e.g., YADL 50k in Figure 7), since unrelated tables may still feature high containment: this problem may occur if the data lake includes multiple variants of the same table, a setting commonly occurring in industry (Halevy et al., 2016). An additional drawback of Jaccard containment is that it does not consider cardinality, so that binary tables would be considered as perfect matches while being useless in practice.

**MinHashLSHEnsemble (MinHash)**  *MinHash* (Zhu et al., 2016) relies on Locality Sensitive Hashing (LSH) to build an index subject to a user-set minimum containment threshold. At query time, all candidate columns with estimated containment larger than the threshold are returned. For consistency with the other methods, we select $k$ candidates from this pool.

The main drawback of MinHash is that it does not feature an inherent ordering of candidates. Moreover, since MinHash returns an approximate result, it may happen that candidates with an actual containment lower than the threshold are returned as False Positives. Under tight constraints, no ordering and False Positives reduce the likelihood of retrieving good candidates (Figure 16). Finally, while the index can be updated when new data is added to the data lake, performance may deteriorate after a certain level of updates, especially depending on the skew of the new data.

**Hybrid MinHash**  To address Minhash's lack of a candidate ranking, we propose Hybrid MinHash. *Hybrid MinHash* leverages the prebuilt index from MinHash to find candidates for a given query column, then uses Exact Matching to rank all candidates. The "top-$k$" candidates are then selected from the ranked list. By measuring the containment over the MinHash query result, the pool of candidates is reduced compared to testing all columns in the data lake (like with Exact Matching).

Just as it takes some of the strong suits of both MinHash and Exact, Hybrid MinHash also shares some of their disadvantages: 1) The MinHash index will degrade each time it is updated. 2) As the first filtering relies on the MinHash results, False Positives increase cost and any candidate missed by MinHash is lost. 3) Re-ranking candidates by calculating the exact containment increases the query time substantially with respect to MinHash (Table 10).

**Starmie**  We compare the simple methods described so far against Starmie (Fan et al., 2023b), a SOTA system for table unionability and join discovery. Starmie employs a contrastive learning method to train column encoders from pre-trained language models to capture contextual information within tables. Starmie performs candidate retrieval by building embeddings for each column in the data lake and for the query column; it then ranks candidates according to the equation:

$$S_{Starmie} \; = \; |Q \cap C_k^i| \cdot S_C(vec(Q), vec(C_k^i))$$

where $S_C$ is the cosine similarity between the embeddings of the query column $Q$ and candidate $C_k^i$.

## B.2 Candidate selection

Each selector receives as input a pool of "$K$" candidates from a given retrieval method on a given data lake, the train and test splits of the base table, and the aggregation method to use. The train split is further split into a training (0.8) and validation set (0.2). The model is trained on the final *integrated* table (i.e., the base table joined with the candidates) produced by the selector of this training split, then the result is scored with the test split.

**Highest Containment Join**  We first consider a simple method that assumes that no ranking is provided in the previous step. *Highest Containment Join* ranks candidates by exact containment, after measuring it for each candidate. The top-1 candidate is joined; ties are broken by taking one candidate at random. This selector re-ranks candidates by containment even if they have already been ranked by the retrieval method, thus it effectively ignores the previous ranking. Due to its simplicity, Highest Containment Join is very quick to execute, but it is likely to select suboptimal candidates.

**Best Single Join**  *Best Single Join* iterates over each candidate one at a time. At each step, it joins the base table on the current candidate, performs the aggregation, trains a ML model, then evaluates the model on a held-out validation step. After all candidates have been evaluated, the best candidate and associated ML model are used: the ML model is re-trained over the entire training split and selected for the final evaluation. The Best Single Join is an exhaustive and expensive method.

**Full Join**  The *Full Join* selector does not include any logic for selecting candidates. Given the base table, the Full Join first aggregates candidates, then joins all of them on the base table. Depending on the budget that is provided and the number of columns in the candidates, this may increase the final number of features in the augmented table by a lot. In most of our experiments, we fix the number of candidates to 30. After joining, the ML model is trained over the fully augmented table. Since training is performed only once, this method is faster than the exhaustive selection methods (Best Single Join and Stepwise Greedy Join), but the large number of resulting features may increase the memory pressure.

**Stepwise Greedy Join**  *Stepwise Greedy Join* is an extension of Best Single Join that aims to add more information than what is available in a single table, while reducing the amount of noise added by joining irrelevant tables. Firstly, the baseline performance is measured by training a model on the base table's training split and validating on the validation split. The base table then becomes the "current table" at the first step of algorithm. Then, candidate selection is performed. Candidates are initially re-ranked by their containment, then each candidate is evaluated sequentially. In each iteration, a candidate is joined on the current table, evaluated on the validation split, then depending on whether it improves performance or not it may be kept or discarded. Any time a candidate is retained, its features will be added to the "current table". Candidates that do not improve performance are discarded. The training procedure of Stepwise Greedy Join is expensive, as the training operation is repeated multiple times, and slows down as the number of features

increases. It has however the advantage of avoiding the addition of suboptimal features. Algorithm 1 reports the steps required.

---

**Algorithm 1** Pseudocode of the Stepwise Greedy Join selector.

---

**Require:** base table $T$, candidate set $\mathcal{C} = \{c_1, c_2 \ldots c_n\}$, scoring function $f$
**Require:** chosen candidates set $\mathcal{S} = \emptyset$
**Require:** predictor ML
  $T_{train}, T_{valid} = \text{get\_training\_splits}(T)$
  $\text{ML.fit}(T_{train})$
  score$=f(\text{ML.predict}(T_{valid}))$
  $t^c_{train} = T_{train}$
  $t^c_{valid} = T_{valid}$
  $\mathcal{C} = \text{rerank}(\mathcal{C})$
  **for all** $i$ in $n$ **do**
    aggregate$(c_i)$
    $t^i_{train} = t^c_{train}$ LEFT JOIN $c_i$
    $t^i_{valid} = t^c_{valid}$ LEFT JOIN $c_i$
    $\text{ML.fit}(t^i_{train})$
    $score_i=f(\text{ML.predict}(t^i_{valid}))$
    **if** $score_i > score_c$ **then**
      $score_c = score_i$
      $t^c_{train} = t^i_{train}$
      $t^c_{valid} = t^i_{valid}$
      $\mathcal{S} = \mathcal{S} \cup \{c_i\}$
    **else**
      discard$(c_i)$
  $t = T$
  **for all** $s_i \in \mathcal{S}$ **do**
    $t = t$ LEFT JOIN $s_i$
  $\text{ML.fit}(t)$

---

**Single-table selectors and multi-table selectors** Highest Containment and Best Single Join produce smaller integrated tables as they only join *one* candidate, rather than *all* potential candidates like Full Join and Stepwise Greedy Join: we can therefore classify the two pairs as **single-table selectors** and **multi-table selectors** respectively. Top-$k$ Full Join can be assigned to either class depending on the value of $k$.

In all selectors that involve a join, aggregation is carried out prior to executing the join itself. In our pipeline, aggregation is carried out before executing any join by grouping the "right table" by the join key, then applying one of the aggregation functions described above. This is to avoid materializing large joins, which have a huge cost in memory and time. Due to how aggregation is carried out, the final result is the same before and after the join, so executing it before materializing the joined table is more efficient.

All join selectors follow the fit-predict paradigm proposed by scikit-learn (Pedregosa et al., 2011), which simplifies extending the pipeline with new selectors.

## B.3 Aggregation

When joining tables for a downstream machine learning task, one-to-many relationships must be aggregated to avoid replicating samples in the base table which would modify the initial data sampling.

Following the example in Figure 6, a join on "Title" leads to the duplication of the two first rows in the base table, as both title values appear in two rows of the candidate table. How should we aggregate the rows to obtain the best prediction over the column "Revenue"? We test three different aggregation strategies:

1. **Any** selects one row from each group, without considering the order.

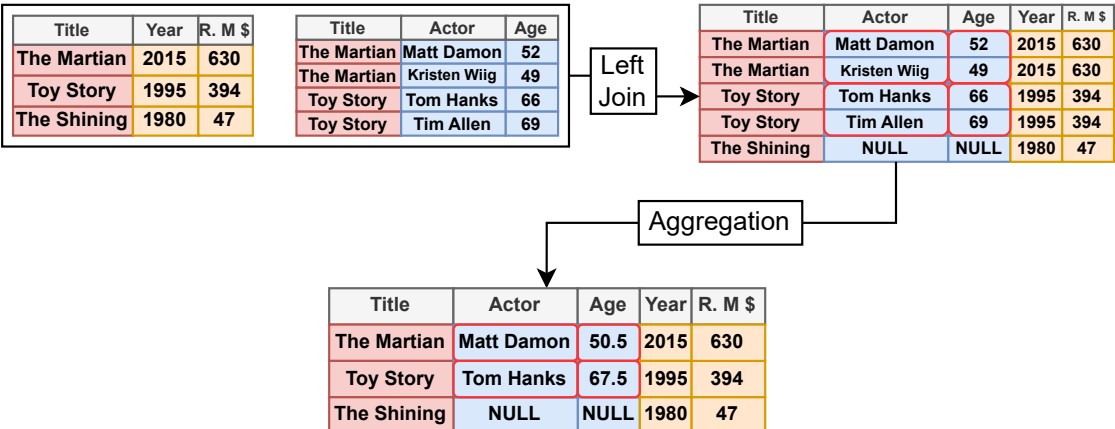

Figure 6: Example of how a left join would duplicate rows from the base table, and how aggregation keeps the same number of samples as the base table (top left).

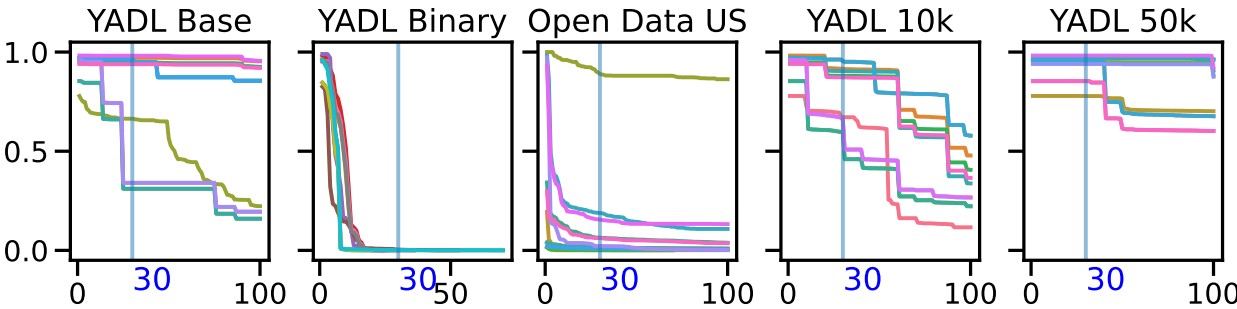

Figure 7: Exact containment measured over the top 100 join candidates for each base table (represented by a colored line) on each data lake. The x-axis reports the rank, rank 30 is the retrieval cutoff we use in our experiments.

2. **Mean** replaces all numerical (categorical) values by the mean (most frequent) of all values for that attribute.
3. **DFS** (Deep Feature Synthesis (Kanter & Veeramachaneni, 2015)) greedily adds new features by performing different aggregations (mean, median, count, standard deviation, etc.) over each column in the table.

## C   Experimental results

### C.1   Choosing which baselines to use

Including recent publications often turned out to be very difficult, as the code was either unavailable, or not suited to extensive experiments on a data lake of the size that we investigate in our work. We worked on including Starmie (Fan et al., 2023b), and (partially) Lazo (Fernandez et al., 2019). We were able to test Starmie on some of the data lakes (Binary, YADL Base, YADL 10k) though we had to adapt and optimize certain sections of the discovery code that did not scale well to our environment. We were unable to run Starmie on the larger YADL 50k, and the less structured Open Data US. Starmie results are illustrated in Section 5. The modified Starmie repository is available here: `https://github.com/rcap107/starmie`, and the improvement was merged in the main repository. These modifications are code optimizations and fixes for corner cases; they do not change the logic of Starmie.

Table 6: Recorded peak RAM usage for building and querying the different retrieval methods. All sizes are reported in MB. Runs that failed are marked as "NA".

| Case | Data Lake Version | Exact Matching | Minhash | Minhash Hybrid | Starmie |
|------|------|------|------|------|------|
| Disk | Binary | 0.0 | 1.4 | 1.4 | 501.1 |
| | Open Data US | 12.7 | 434 | 434 | NA |
| | YADL 10k | 21.3 | 515 | 515 | 522.3 |
| | YADL 50k | 28.2 | 2480 | 2480 | NA |
| | YADL base | 21.3 | 462 | 462 | 522.4 |
| RAM Build | Binary | 1196 | 226 | 226 | 19424 |
| | Open Data US | 5466 | 3288 | 3286 | NA |
| | YADL 10k | 688 | 3312 | 3290 | 87716 |
| | YADL 50k | 1259 | 13697 | 13697 | NA |
| | YADL base | 1061 | 2810 | 2810 | 38683 |
| RAM Query | Binary | 255 | 216 | 216 | 4567 |
| | Open Data US | 384 | 7789 | 10472 | NA |
| | YADL 10k | 388 | 9138 | 12679 | 134869 |
| | YADL 50k | 781 | 37332 | 37327 | NA |
| | YADL base | 412 | 6784 | 8541 | 145437 |

Lazo relies on a client-server architecture that requires Elasticsearch v7. We were able to set up the backend environment and integrate the Lazo client in our pipeline, but we are unable to execute querying operations on our data lakes because the client database crashes due to out of memory errors. The wrappers we developed for testing Lazo are available in the main repository (`https://github.com/rcap107/retrieve-merge-predict`).

Additionally, for the retrieval step we considered ALITE (Khatiwada et al., 2022), KOIOS (Mundra et al., 2023), JOSIE (Zhu et al., 2019), and Saibot (Huang et al., 2023a). Each method requires substantial changes to the original codebase or very specific build configurations to run on datasets and environments different from those included in the original paper.

For the prediction pipeline, we consider ARDA (Chepurko et al., 2020) and Metam (Galhotra et al., 2023), focusing on Metam as it builds upon ARDA. Similarly to the retrieval case, Metam required substantial changes to be executed on data other than that in the paper and when tested in our environment it did not produce augmented tables. We implement some of the solutions suggested by Metam in our Stepwise Greedy Join selector, and we re-use the "Open Data US" data lake and "schools" base table in our experimental section. The modifications we made to the Metam code to interact with our pipeline are available at `https://github.com/rcap107/metam`.

We were not able to find an open repository for SilkMoth (Deng et al., 2017b), DeepJoin (Dong et al., 2023), Mileena (Huang et al., 2023b), or Kitana (Huang et al., 2023c).

## C.2 Experimental setup

We run our experimental campaign on a SLURM cluster, fixing the number of threads to 32. Nodes have at least 256GB of RAM. Experiments that involved NNs were run on nodes equipped with GPUs. Overall, preparing the retrieval methods required about 1050 CPU hours, while the experimental results required about 189k compute hours (about 21 years of equivalent CPU time); CatBoost and RidgeCV were run on CPU, ResNet and RealMLP on GPU. Table 7 reports the equivalent CPU (GPU) time across all experiments, broken by ML model; running experiments with 32 CPUs would reduce this runtime by about 32x.

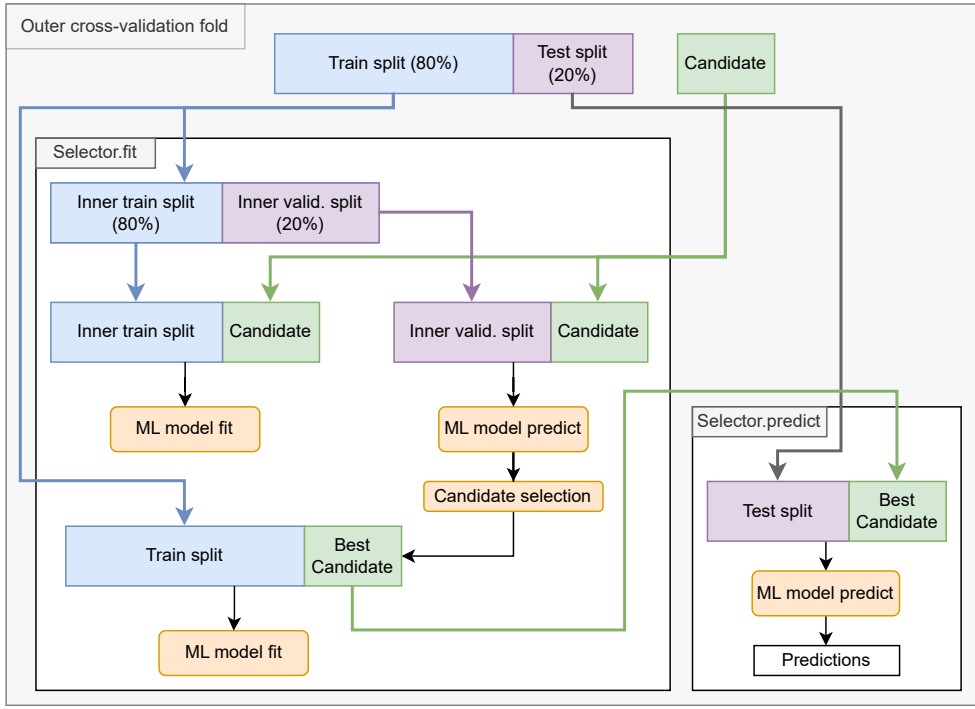

Figure 8: Schema of the cross-validation setup used in the training pipeline.

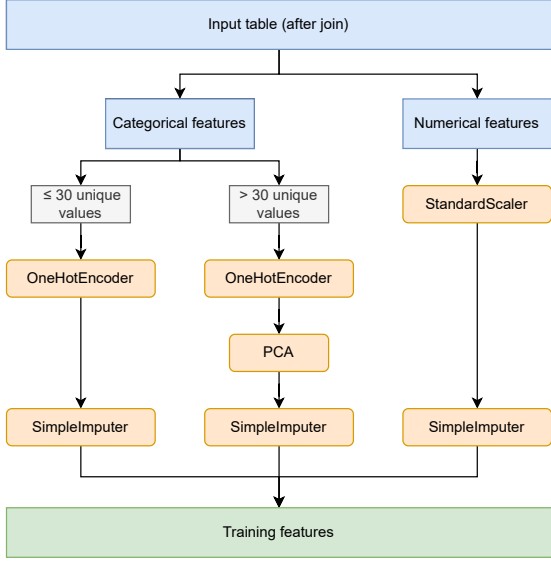

Figure 9: Pre-processing steps applied to input tables before being fed to the parametric ML algorithms. Catboost does not require preprocessing. The names reported here reflect the corresponding scikit-learn (Pedregosa et al., 2011) objects.

**Implementation details** The implementation of the pipeline is in Python[7]; Exact Matching, aggregation and join operations are implemented using Polars (Vink et al., 2024) as backend. For the Prediction step, RidgeCV is implemented with Scikit-learn (Pedregosa et al., 2011), CatBoost using the official library (Prokhorenkova et al., 2018), ResNet and RealMLP use pytabkit (Holzmüller et al., 2024).

We rely on the Python implementation of MinHash provided by the Datasketch package (Zhu et al., 2023). For Starmie, we optimize the original implementation so it can scale to work in our environment. We use the official DFS package. Data structures are stored by persisting on disk the ensembles for MinHash, the candidate ranking for each query column for Exact Matching, and the model checkpoints for Starmie; Hybrid MinHash relies on the MinHash data structures to work, so no additional storage is required for it.

Table 6 reports the size on disk of the data structures used for each method.

Table 7: **Total equivalent compute hours, days, months and years required to run all the experiments. Single-CPU or single-GPU equivalent time: having a 32-CPU computer divides the time by 32.**

| Predictor | Platform | Total compute time |
|---|---|---|
| RidgeCV | CPU | 4y 3m 10d 7h |
| CatBoost | CPU | 1y 3m 29d 21h |
| ResNet | GPU | 5y 6m 23d 0h |
| RealMLP | GPU | 10y 7m 23d 3h |
| Total | Both | 21y 9m 26d 8h |

**Retrieval** We use a containment threshold of 0.2 for the preparation of the MinHash index, and clamp the number of candidates returned by each retrieval method to 30 (except when specified otherwise). These values were chosen to balance execution time and expected number of candidates given the distribution of containment encountered in the different data lakes (Figure 7). For Starmie, we use the default parameters defined in the original repository.

**Selection** We fix the number of Stepwise Greedy Join iterations to 30: this number is consistent with the number of candidates that are provided in the retrieval step. None of the other join selectors have parameters to tweak.

**Prediction** We fix the number of CatBoost iterations to 300; we stop training the model 10 iterations after the optimal metric has been detected; we set the L2 regularization coefficient to 0.01. As we are interested in evaluating the effect of each pipeline task rather than optimizing the prediction performance, we do not apply HPO to CatBoost.

We use the default parameters for RidgeCV as used in the scikit-learn implementation.

For RealMLP and ResNet we use the parameters that are set in the pytabkit package as they have been shown in Holzmüller et al. (2024) to be the "better defaults".

### C.3 Critical difference plot

We use critical difference plots to represent an overall ranking of the downstream prediction performance of all the configurations we considered. Specifically, the prediction metric is averaged over all base tables and data lakes, and the resulting value is used to rank each configuration.

Similarly to Figure 2 (and other Pareto plots), we present the same results after splitting them in retrieval method (Figure 10), join selector (Figure 11), and ML model (Figure 12).

### C.4 Aggregated results

Figure 13 is prepared like Figure 2, however this version includes results from all data lakes and does not include Starmie as it could not run on Open Data and YADL 50k. Results are quite consistent with Figure 2: this means that the new data lakes (YADL 50k and Open Data US) do not alter the overall trends much.

In Figure 14 we report the fold vs fold difference in prediction score ($R^2$ and AUC) and relative execution time with respect to the reference configuration (Exact Matching, Best Single Join, aggregation Any and

---

[7]Repository: `https://github.com/rcap107/retrieve-merge-predict`

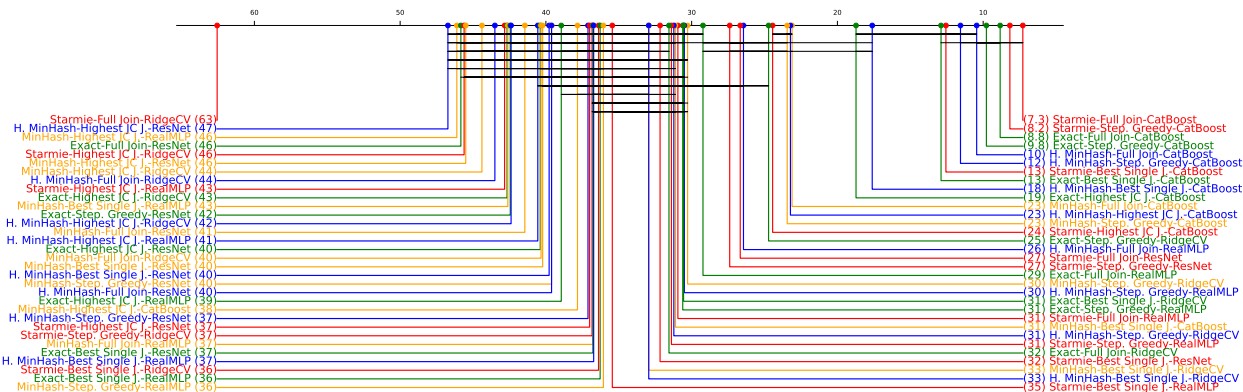

Figure 10: Critical difference plot for the retrieval methods: Exact Matching, MinHash, Starmie, Hybrid MinHash.

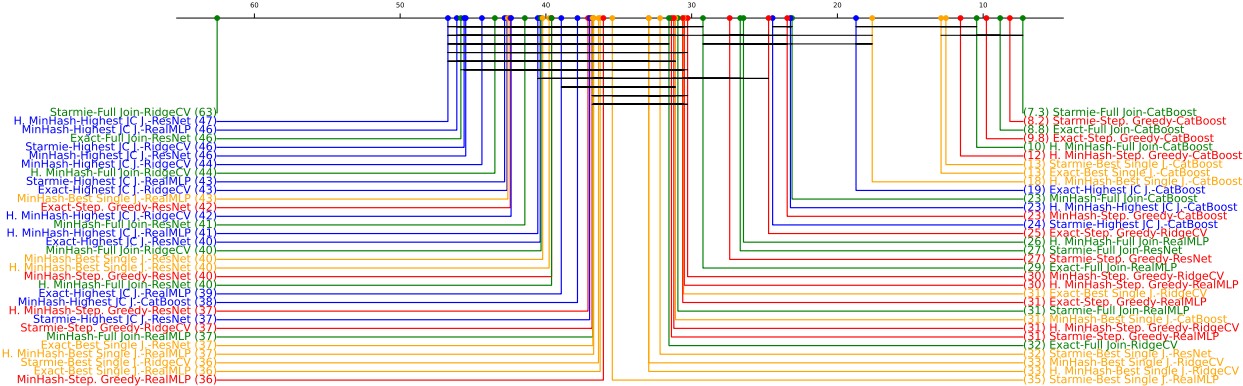

Figure 11: Critical difference plot for the join selectors: Full Join, Stepwise Greedy Join, Highest Containment Join, Best Single Join.

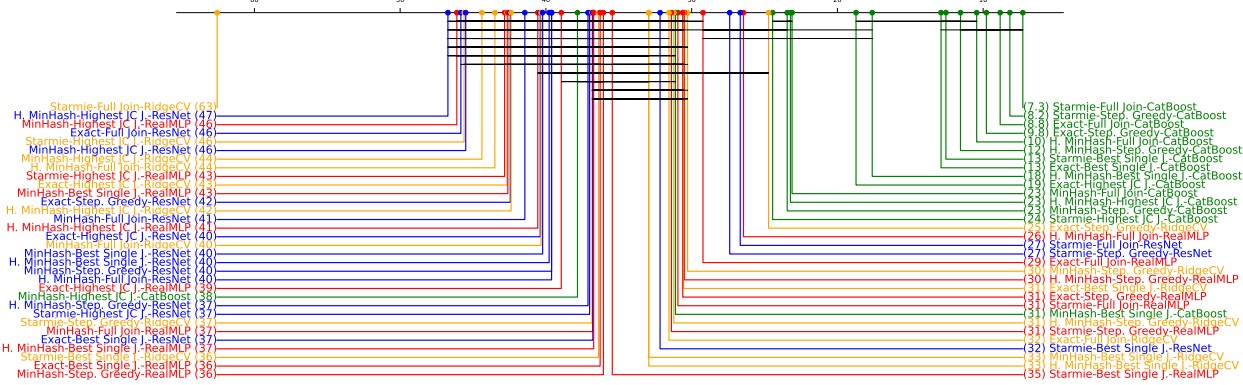

Figure 12: Critical difference plot for the predictors: CatBoost, RidgeCV, RealMLP, ResNet.

Table 8: **Aggregated prediction results broken by base table and data lake**. Starmie results are not reported here because it could not run on Open Data US and YADL 50k. As many ResNet and RidgeCV runs did not converge, we report a trimmed mean with cutoff 0.2.

| Base Table | Binary | YADL Base | YADL 10k | YADL 50k | Open Data US |
|---|---|---|---|---|---|
| Company Employees | $0.086 \pm 0.036$ | $0.081 \pm 0.043$ | $0.071 \pm 0.052$ | $0.113 \pm 0.048$ | $0.011 \pm 0.019$ |
| Housing Prices | $0.16 \pm 0.068$ | $0.187 \pm 0.077$ | $0.143 \pm 0.074$ | $0.199 \pm 0.092$ | $0.221 \pm 0.09$ |
| Schools | NA | NA | NA | NA | $0.892 \pm 0.137$ |
| 2021 US Accidents | $0.138 \pm 0.065$ | $0.178 \pm 0.054$ | $0.199 \pm 0.076$ | $0.150 \pm 0.058$ | $0.229 \pm 0.028$ |
| US County Population | $0.725 \pm 0.203$ | $0.078 \pm 0.064$ | $0.075 \pm 0.074$ | $0.112 \pm 0.148$ | NA |
| 2020 US Elections | $0.374 \pm 0.062$ | $0.373 \pm 0.115$ | $0.358 \pm 0.054$ | $0.394 \pm 0.045$ | $0.400 \pm 0.06$ |

CatBoost) for retrieval (Figure 14 (a)), selection (Figure 14b), aggregation (Figure 14c), and prediction ( Figure 14d). Individual folds are reported as dots; color palettes depend on the data lake. This is an alternative way of representing the data in Table 2. Note that as we are comparing folds against folds, the difference is 0 when a particular parameter is the same as the reference. We choose Best Single Join as a reference for the selector because DFS could not run with Full Join and Stepwise Greedy Join.

**Retrieval** Figure 14(a) shows that Exact Matching and Starmie have very similar performance (a median difference of 0.06%), with Starmie outperforming Exact Matching in some instances (this can also be observed in Figure 2 and Figure 10), which makes sense as the similarity metric used by Starmie combines Jaccard similarity with the cosine similarity of column embeddings. Base MinHash performs very poorly, with a median difference of -17.48% with respect to Exact Matching. This is not surprising and is likely caused by the lack of a candidate ranking combined with the presence of a candidate budget. Hybrid MinHash shows a marked improvement over base MinHash, gaining about 14% in median: this confirms that it is an effective strategy to address some of the shortcomings of the base method.

MinHash is faster than the others, despite the fact that the candidate budget $k$ is 30 for all methods. This is because, on average, the candidates retrieved by MinHash have a much lower containment than those proposed by the other methods (Figure 3a): due to the smaller amount of data to move and use for training

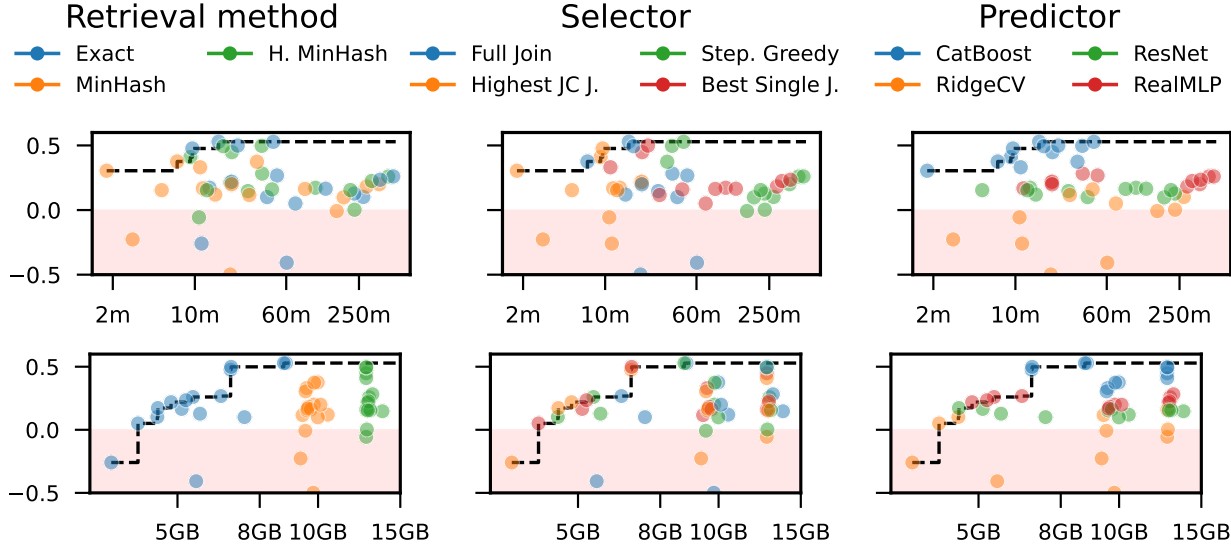

Figure 13: **Pareto frontier analysis for the pipeline steps on all data lakes.** The prediction performance ($y$-axis) is plotted against retrieval + run time (top) and peak RAM usage (bottom). Time for offline retrieval preparation for MinHash is not reported here. This figure includes all data lakes, but no results with Starmie.

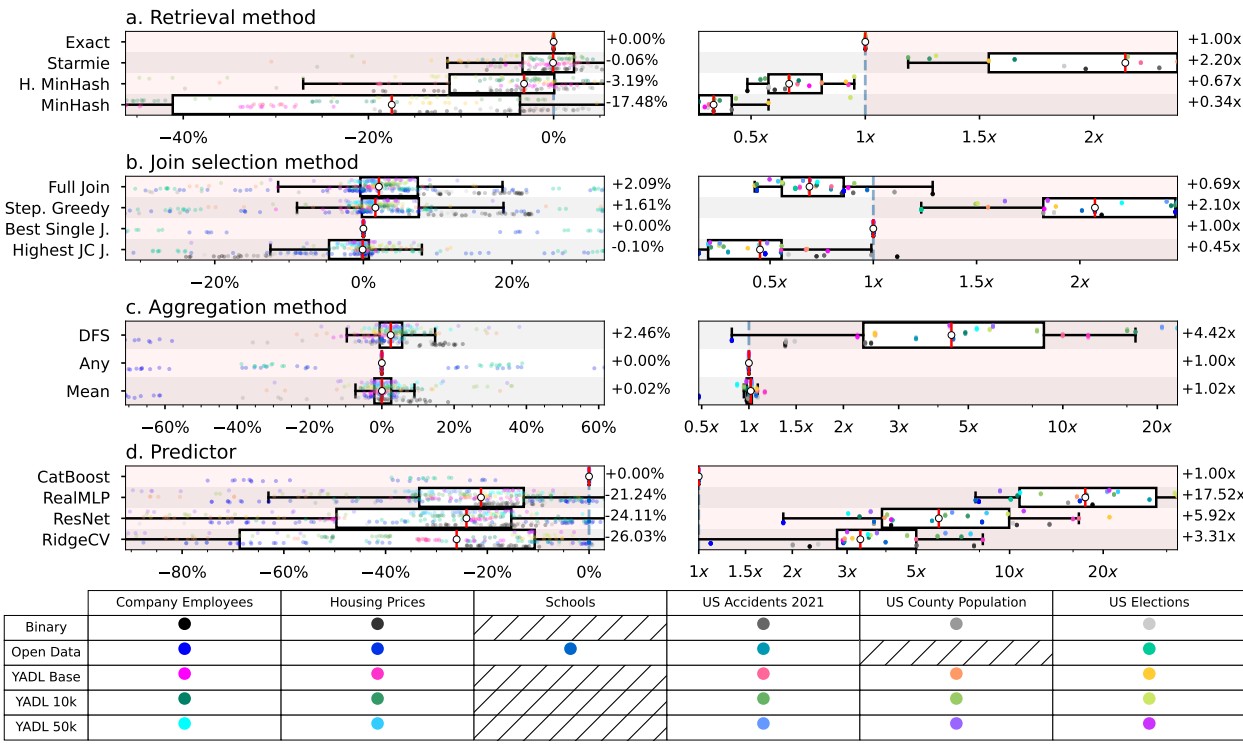

Figure 14: Experimental results across all data lakes, comparing the performance difference between evaluated methods in different steps of the pipeline. The median difference is reported on the right of each plot. (a) compares join selectors, (b) aggregation methods, and (c) ML models. In all cases, results are relative to the "average method".

the models, the runtime is shorter; furthermore, imperfect recall affects the performance of both MinHash and its hybrid variant.

While Hybrid MinHash appears to be faster than Exact Matching in the pipeline, it is important to note that re-ranking candidates incurs a non-negligible additional cost (Figure 14(a), Table 10).

Overall, the two methods based on precise ranking (Exact Matching and Starmie) outperform the methods based in approximate matching (MinHash and Hybrid MinHash), suggesting that the computational cost incurred in measuring exact containment does result in better prediction performance.

**Selection**   The choice of selector shows a clear effect when moving from single-table selectors (Highest Containment and Best Single Join) to multi-table selectors (Stepwise Greedy and Full Join), bringing a benefit of up to 2.2% in median (Figure 14(b)).

Stepwise Greedy Join is much slower compared to all other methods. This is not surprising, since this selector executes a join and trains a model in each iteration, then re-trains the model at the end of the fit step. In practice, while this added complexity brings an improvement of about 1.6% with respect to Best Single Join, this comes at the cost of a 2.1x increase in run time; even worse, this is not enough to beat Full Join in all cases, as the latter method is both faster and slightly better in prediction performance. Highest Containment is significantly faster than all other solutions because it only re-ranks candidates before joining the top-1 candidate.

The difference in performance between the two single-table selectors (Best Single Join and Highest Containment) is likely due to the fact that, while Jaccard Containment is an indicator of a potentially good join, it is not sufficient for selecting the best candidate. For example, multiple candidates may have the same containment value, leading to ties. Given the lack of better information, Highest Containment breaks ties at random, thus it may select tables that are not relevant. Redundant data lakes like YADL 50k (Figure 7) are more affected by this problem. The significant difference between single- and multi-table selectors is explained by the learning model benefiting from an increased set of features: merging more than one table inherently results in a richer feature set.

It is important to mention that the picture is more complex than what can be observed exclusively from Figure 14. In fact, Figures 19 and 20 show that, while Full Join appears to be better on average, there are configurations of hyperparameters in which it performs worse than Stepwise Greedy Join due to the addition of unrelated features.

A final important observation is that all selectors rely on the candidates proposed by a retrieval method: if these candidates have poor quality, the selectors cannot compensate for that.

**Aggregation**   Aggregation experiments do not include Full Join and Stepwise Greedy Join because DFS ran out of resources with those selectors. In fact, while DFS brings some benefit in its generation of new features (up to 2.46% in median wrt Any), it is also extremely slow (up to 4.42x wrt the simpler method), and this was in a simpler case where only one table was joined at a time. With multi-table selectors, this problem became even more noticeable and prevented us from testing the multi-table selectors. This very large memory cost is consistent with Cvetkov-Iliev et al. (2023).

For what concerns the simpler methods, Mean is slightly better than Any, and has a slightly longer runtime; we did not observe major differences between the two methods in practice.

**Predictor**   The performance difference between CatBoost and all other methods already highlighted in Figure 2 and Figure 12 is confirmed in Figure 14(d), which shows CatBoost outperforming the second best model in RealMLP by about 21.2%, and the worse NN model in RidgeCV by 26%. Interestingly, CatBoost is also *faster* than all other methods by a significant amount, likely owing to the optimized preprocessing of categorical variables. RealMLP is the better performing NN model, and has shown a good resilience in the very challenging environment that we are considering; however, it is much slower than all other methods, even when run on GPUs, requiring up to 17x as much time as CatBoost.

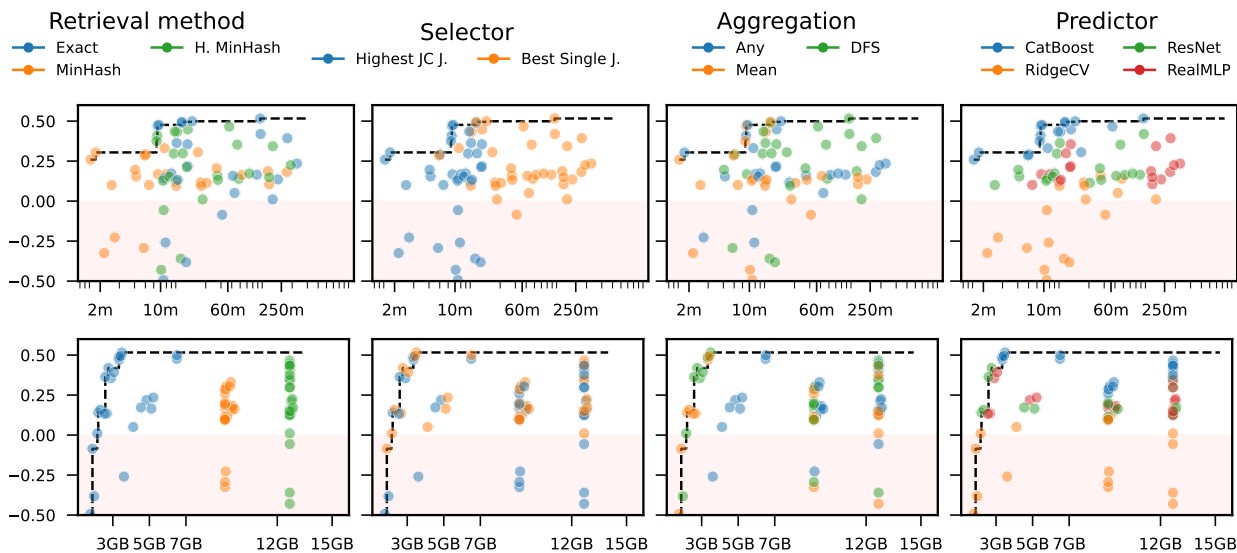

Figure 15: **Pareto frontier analysis testing different aggregation methods.** The prediction performance ($y$-axis) is plotted against retrieval + run time (top) and peak RAM usage (bottom). Each row presents the same results, broken down by retrieval method (left), join selector method (center) and predictor (right). Starmie, Full Join and Stepwise Greedy Join are not reported here.

**Different Lakes** Table 8 reports the trimmed average of the prediction performance across all configurations, broken by base table and data lake. We observe experimentally that many RidgeCV and ResNet runs do not converge, leading to very long run times and extremely large negative values for $R^2$. For this reason, we use a trimmed mean with a cutoff of 0.2. Indeed, the results are so poor because the mean across predictors is lowered substantially by the performance of the parametric models. Internal tables could were only run on the data lake they were sampled from, and other cells are reported as "NA".

Despite its lower containment compared to YADL-based data lakes (Figure 3a and Figure 7), Open Data US shows good results, outperforming the YADL variants in some cases: this may be due to the smaller fraction of nulls, and larger tables on average compared to YADL.

Prediction performance was particularly good on US demographic-adjacent tables, and on the internal dataset "Schools", which achieved perfect classification performance in most cases – also visible in Figure 3b, where highest containment leads to best performance.

YADL's internal table (US County Population) exhibits a similar, though not quite as extreme, behavior to that of Schools. This can be explained by the fact that internal tables, i.e., tables that are sampled from the data lake itself, tend to have copies in the data lake, which are likely to contain information that is correlated with the prediction task. This highlights the specific behavior of internal tables, which are routinely used in the experimental campaigns in the literature.

The different prediction performance obtained with each YADL variant suggest that the structure of each data lake impacts the actual prediction performance.

## C.5 Smart aggregation brings some benefit at a major cost

While containment impacts heavily both prediction performance and execution time, aggregation has a similar impact on the execution time, without the same degree of improvement for the prediction performance (Figure 14c).

Aggregating values always leads to a loss of information: in exchange for a larger cost, complex aggregation methods that preserve more information (DFS) or that replace values with better representatives of a group

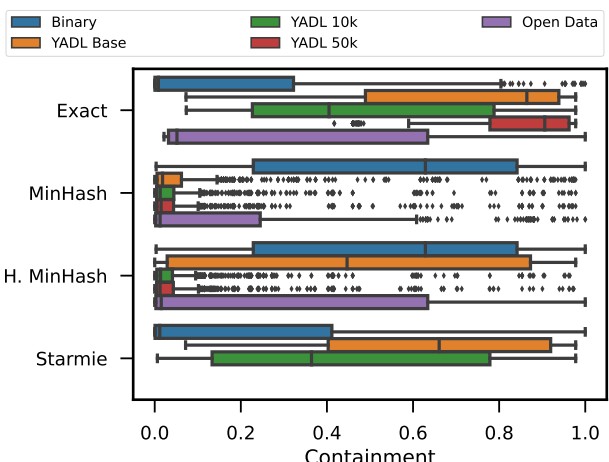

Figure 16: Distribution of containment in the top-200 candidates obtained by each retrieval method on different data lakes.

(mean) should lead to better prediction performance. To an extent, this is what we observe: DFS does improve the downstream performance by a decent amount, however this is at a major computational cost. This problem is exacerbated by the fact that aggregation must be performed whenever a join is executed *at any point in the pipeline*, including joins executed during join selection. The result is a compounding slow-down of the entire pipeline.

The similar performance of "any" and "mean" can be explained by the execution of joins on key-like columns so that values in these columns tend to be mostly distinct, thus producing the same aggregated value for both "mean" and "any". The slight difference in performance and runtime between the two methods may be attributed to the large fraction of missing values in some of the data lakes: the "mean" method is biased towards selecting the most frequent value, which may be a null. When that happens, more values than needed become nulls, reducing the amount of features and introducing a slight speedup in the overall training.

The relatively unsatisfying performance of DFS is also explained by the fact that we are not fully exploiting its capabilities. We consider only join depth-1 chains: at each aggregation step, we join the base table with an additional table, rather than leveraging the recursive generation of features provided by DFS. As a result, DFS is not as effective at generating features as it would be with deeper join paths (Cvetkov-Iliev et al., 2023) and may benefit from integrating join path discovery systems (Deng et al., 2017a).

Figure 15 confirms that, while DFS improves the prediction performance, its overall cost remains problematic.

### C.6   Distribution of the containment for each retrieval method

To understand the success of the different retrieval methods, we look at the containment of the joins that they suggest. Figure 16 reports top-200 containment. It highlights that, even when retrieving a large number (200) of candidates, the average containment of MinHash is very low compared to the other methods: for larger datasets, MinHash returns thousands of candidates of which only a fraction are selected. The lack of an internal ordering means that high-containment candidates are likely to be missed (Section 4). While MinHash has a threshold that can be tweaked to reduce the number of candidates that are retrieved, we observe that recall drops sharply at high thresholds. Hybrid Minhash has some success in mitigating the problem, as it improves the average containment and the overall downstream performance (Figure 2 left). [8] We were unable to obtain the containment results for Starmie on YADL 50k and Open Data US.

---

[8]The performance of MinHash on Binary is an artefact: the method could only retrieve about 6 candidates on average compared to the 30 found by the other two methods, thus values are averaged across fewer candidates with higher containment.

Table 9: Statistics for the base tables.

| Base Table | Target | # Num. Att. | # Cat. Att. | # Rows |
|---|---|---|---|---|
| Company Employees | Number of Employees | 2 | 7 | 3109 |
| Housing Prices | House Price | 3 | 7 | 22250 |
| Movies | Movie Revenue | 8 | 10 | 7397 |
| Schools | School Class | 4 | 3 | 1774 |
| 2021 US Accidents | Accidents by County | 1 | 3 | 14850 |
| US County Population | County Population | 1 | 1 | 3059 |
| 2020 US Election Results | Vote % by County | 3 | 6 | 22093 |

## C.7 Execution time breakdown

Figure 17 breaks down where the time is spent by each join selector, with the corresponding average total time on the right. *Prepare* tracks the time spent building data structures (including any ranking operation), and loading data. *Train(model)* and *Predict(model)* track the time spent inside the ML model for training and prediction, respectively. Finally, *Train(join)* and *Predict(join)* track the time spent executing a merge operation, combining join and aggregation. Results are aggregated over all experiments with "first" as aggregation. The time distribution follows our expectations: most of the time is spent fitting models, with time spent joining (and aggregating) in second place. Highest Containment Join spends a relatively long time in the "Prepare" step due to the need to re-rank candidates before joining: since the join and train steps involve only one table, they are faster in comparison. Stepwise Greedy Join has a similar re-ranking step, however training the models in each iteration dominates the other

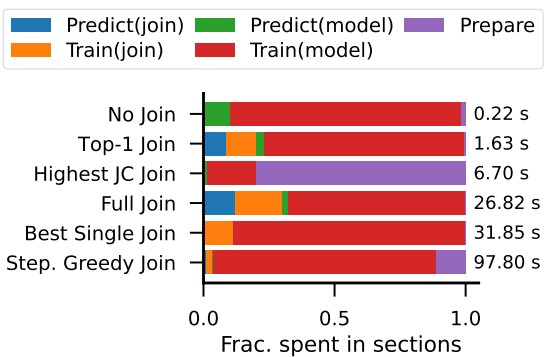

Figure 17: Breakdown of where time is spent for each selector. On the right we report the average run time of each selector.

steps. Full Join merges all candidates at the same time, then trains a single model on the result: this explains how the fraction of time spent joining is larger than in other methods. Top-1 Full Join is a reference of how long it would take to simply join one candidate and train on that, without any additional operation: as expected, it is very fast.

## C.8 Computational trade-offs

Figure 22 compares resources for the different retrieval methods *at indexing and query time*[9]: Starmie is by far the most expensive method, both in run time and in peak RAM. The rapid growth of RAM usage, even for relatively small data lakes (see Appendix Table 6), is particularly problematic and prevents the method from running on GPUs, thus slowing it down even further. The other methods are far cheaper by comparison, and can run on CPU without issues.

Figure 18 shows how different retrieval methods scale with the number of query columns: the value for 0 columns shows the time required to build the index (averaged over all data lakes[10]), and the slope is the

---

[9]We separate retrieval proper from the pipeline execution to clearly highlight their corresponding cost and trade-offs.
[10]Open Data and YADL 50k not included.

time required to query a single column (averaged over all query columns); values are reported in Appendix Table 10. The figure highlights the point at which the cost of recomputing the containment for each new query column on every column in the data lake (Exact Matching) becomes more expensive than building the index (MinHash), and then querying it.

It is clear that, as the number of query columns increases, Exact Matching scales much worse than the other solutions (except Starmie), while Hybrid MinHash and MinHash scale much better: *this is important if the query column is unknown or a user would like to test multiple columns to augment one or more tables.* In any case, Starmie is much slower than all other alternatives.

Figure 18 is prepared assuming that query retrieval time increases linearly with the size of a table and that the cost of creating the MinHash index is fixed for the data lake at hand; we use Table 10 as reference to build the figure. We observe that the MinHash indexing cost pays itself off after as few as three queries on average thanks to the fast query time; Hybrid MinHash requires more queries to break even due to its slower query time, yet it remains faster than Exact Matching when the number of query columns is larger. While the assumptions may not hold in general, the plot gives a reasonable estimate of the break-even points. Starmie scales much worse than other methods as it has a much slower build and query time, and a much larger memory footprint (Figure 22, Table 6).

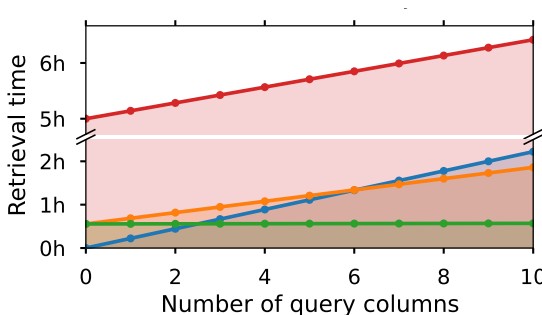

Figure 18: **Runtime as a function of the number of query columns** for the different retrieval methods.

Although dependent on implementation, another factor that should be considered is the size on disk of the indices: the MinHash index occupies a much larger space on disk compared to the data required to hold the Exact matching ranking. Starmie needs to store model checkpoints, which occupy roughly 500MB regardless of the size of the base data lake (Table 6).

Exhaustive computation of the containment is a net gain in performance at the expense of an execution time that increases quickly as the number of columns to query increases. This may not be a problem if the user is aware of which columns should be queried; if, instead, the user is trying to conduct an exhaustive search over all columns, a method such as MinHash should be favored. These observations are consistent with Zhu et al. (2019). Finally, in scenarios where the query table and the data lake do not change, query results can computed offline and reused; in these scenarios, the additional cost of Exact Matching would be less problematic. In situations where the data lake tends to evolve over time, methods that support updating the index such as Fernandez et al. (2019) or Fernandez et al. (2018) should be considered.

### C.9 Interplay between different variables

In Figures 20 and 19 we report the interplay between each variable (Predictor, Retrieval method, Selector, Data lake, Base table) with each other. Each row in a figure is a boxplot that represents a specific variable, while each column on the row breaks down the results by a different variable. We provide these figures to showcase how specific configurations may have behaviors that are hidden by the necessity to aggregate information for the main body. Some interesting takeaways that emerge from this representation are the following:

Table 10: **Average time required to prepare retrieval methods**. Build and query time are separate as index construction can be done offline.

| Method | Avg. Index time | Avg. Query Time |
|---|---|---|
| Exact | - | 826.4 |
| MinHash | 2107 | 2.92 |
| H. MinHash | 2107 | 473.0 |
| Starmie | 18196 | 512.4 |

- RealMLP Holzmüller et al. (2024) is more resilient than all other parametric methods even in the challenging configurations we are considering.
- RidgeCV is even more penalized by larger tables (see the results with Full Join and on Open Data)

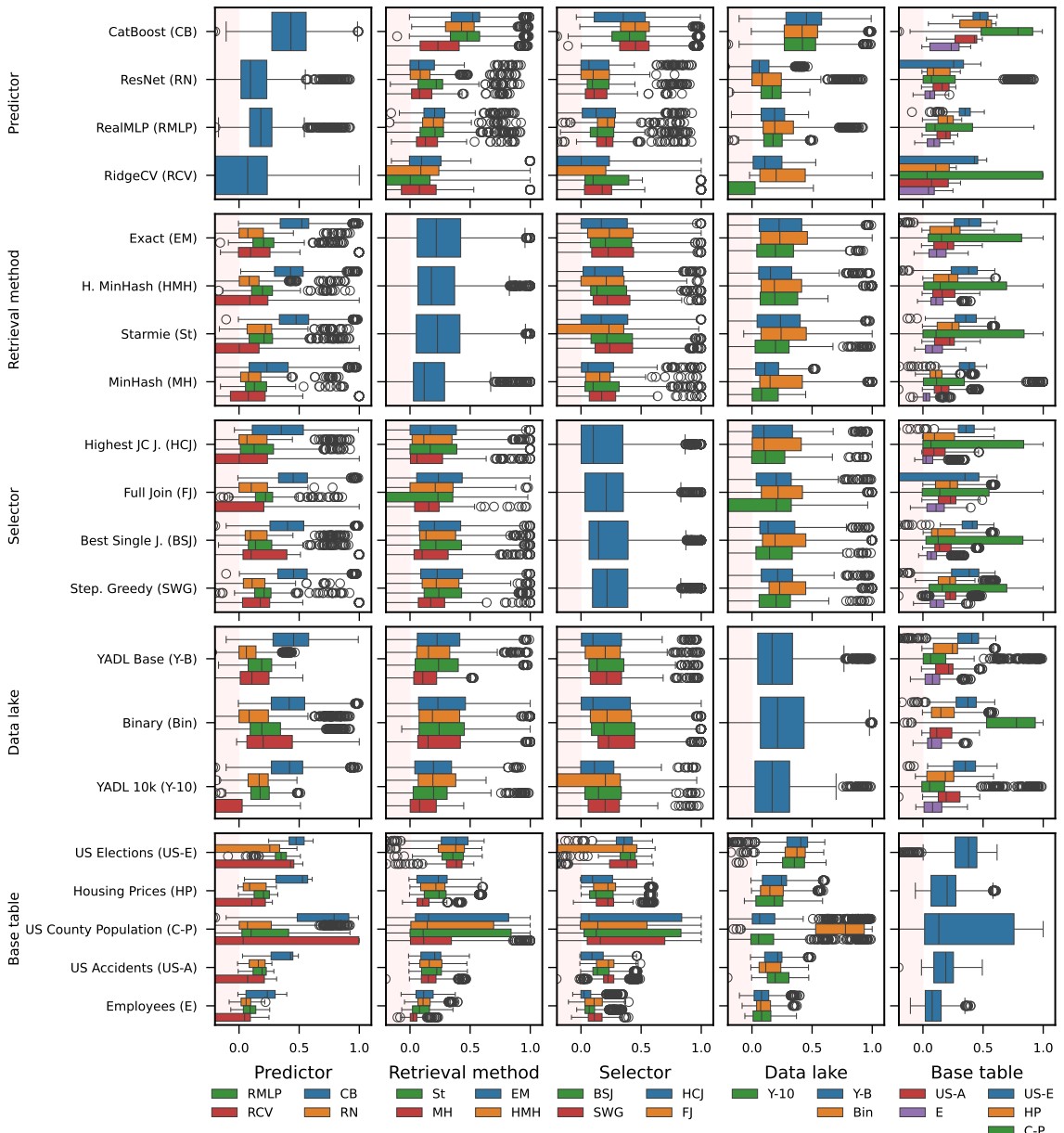

Figure 19: Effect of different variables on the (absolute) prediction performance. Each row reports one variable (e.g., Predictor) and breaks it down by another variable on each column (e.g., by Retrieval method). This plot does not include YADL50k and Open Data US.

- Full Join and Stepwise Greedy Join have similar performance, but Full Join has worse worst-case performance on some data lakes. This confirms the fact that when noise is more prevalent, Full Joining is not a good strategy.
- Overall, Starmie does not bring a lot of benefits and behaves similarly to Exact Matching in most configurations.
- Other than Catboost for predictors, no other method appears to clearly dominate its category.

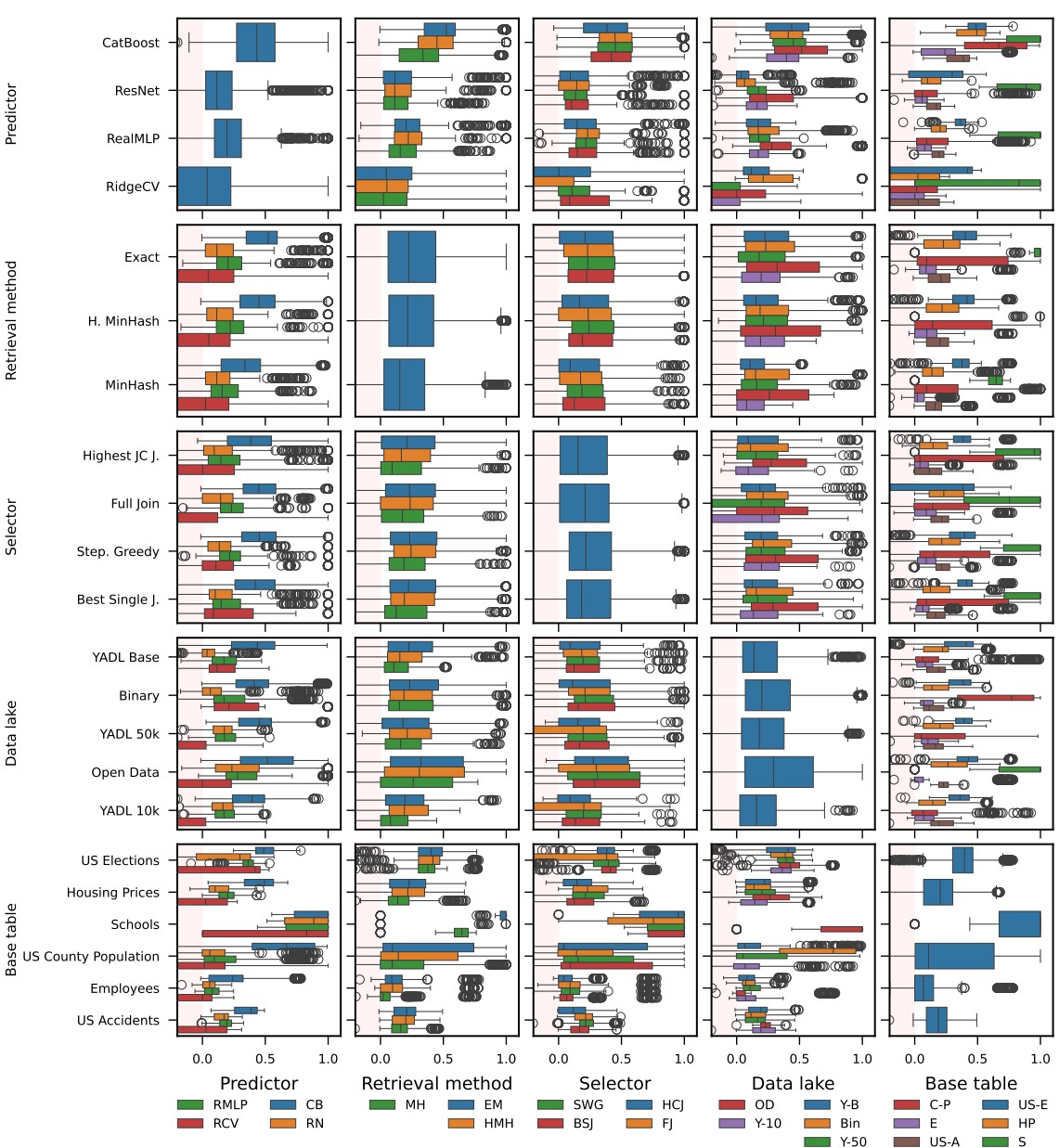

Figure 20: Effect of different variables on the (absolute) prediction performance. Each row reports one variable (e.g., Predictor) and breaks it down by another variable on each column (e.g., by Retrieval method). This plot does not include Starmie.

### C.10   Effect of top-k

In this section we give provide additional context on the impact of $k$ on the pipeline. Figure 21 shows that the prediction performance saturates quickly as the number of candidates increases, while the memory footprint increases sharply: this behavior is similar to the impact on runtime shown in Figure 4.

Figures 23, 24, and 25 show how increasing the value of $k$ affects the prediction performance, fold runtime, and peak RAM usage respectively as a function of the data lake and base table. We use the reference configuration to obtain these results. The data lake has a large impact on the pipeline, with Open Data and YADL10k having a much larger RAM footprint. This may partially be an artefact of how YADL10k is generated; in Open Data's case, this is likely due to the fact that some of the tables in the data lake contain potentially hundreds of columns.

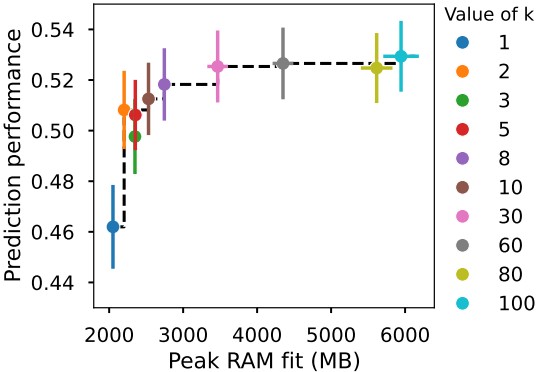

Figure 21: Pareto plot showing how the performance changes as a function of the peak fit RAM for different values of $k$.

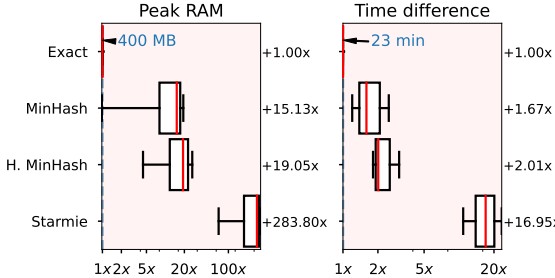

Figure 22: **Resource cost of indexing the retrieval methods**. The computational cost of each retrieval method is plotted with respect to the reference (Exact Matching). The median difference is reported on the right of each plot.

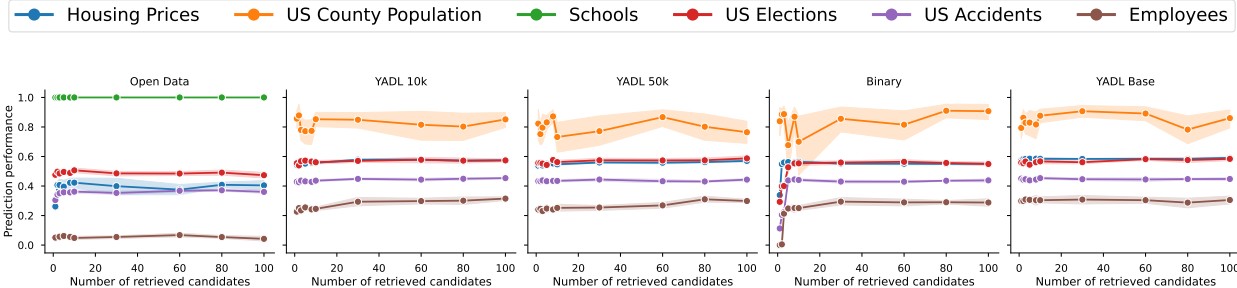

Figure 23: Prediction performance as a function of the number of candidates obtained in the retrieval step, broken by data lake.

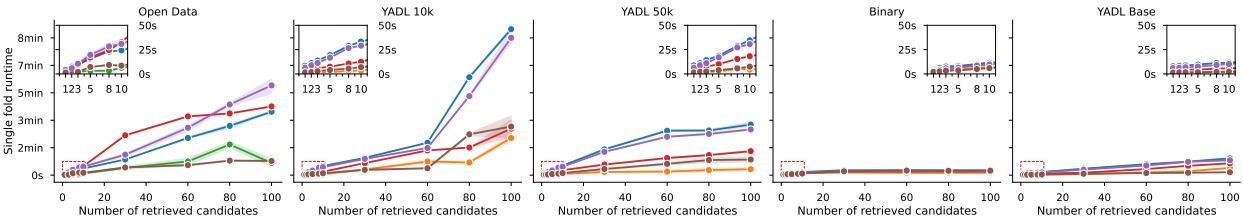

Figure 24: Fold runtime in seconds as a function of the number of candidates obtained in the retrieval step, broken by data lake.

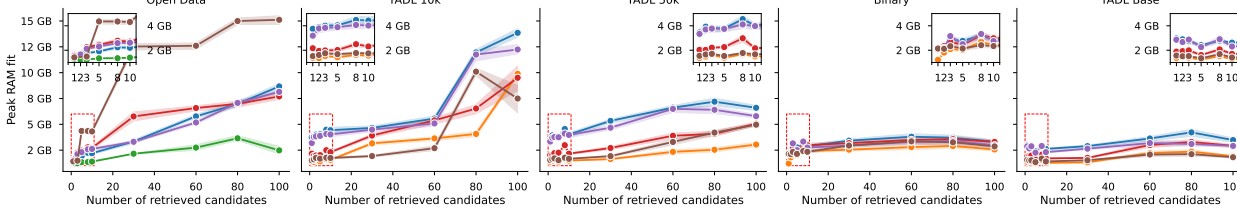

Figure 25: Peak RAM required at fit time as a function of the number of candidates obtained in the retrieval step, broken by data lake.

