# OpenReview forum: "Retrieve, Merge, Predict: Augmenting Tables with Data Lakes"
_TMLR — Accepted by TMLR_

### Review · Reviewer_DvRn · 2025-02-10

**Summary Of Contributions:**

This paper focuses on the complete data augmentation pipeline for modeling that involves starting from a base table, and utilizing a data lake to augment the base table with additional columns for each row in the base table, and finally building a machine learning model on that augmented table. While there has been a lot of work analyzing and automating the choices (such as hyperparameters) in the model building stage, this paper focuses on all the choices in the data augmentation stage which involves retrieval of relevant candidates from the data lakes, and the merging of candidates to create the augmented features. Both these steps involve multiple choices. To this end, this paper
- Presents a benchmark for such pipelines to evaluate the predictive performance/computational cost trade-offs for the different configurations of this pipeline.
- Evaluates various such pipeline configurations on this benchmarking suite to develop valuable insights in terms of the capabilities of the different configurations, and provide areas that could be most beneficial for future research in this setup.

**Audience:**

Yes

**Broader Impact Concerns:**

No broader impact concerns

**Claims And Evidence:**

Yes

**Requested Changes:**

I think the presentation of the results can be enhanced to address some of the questions I raised in the Weaknesses above. Maybe the use of relative metrics (before the aggregation across all base tables) can also improve presentation.

Beyond this, I have a couple of questions:
- **(Q1)** Only considering methods based on Jaccard similarity will of course lead us to a scenario where the join candidates are selected agnostic to the downstream tasks. I think the embedding based matching scheme can incorporate some relevance to downstream tasks. As fuzzy matching is not considered in this work, it is important to discuss this aspect more explicitly to highlight the caveats of the insights formed from the elaborate evaluation.
- **(Q2)** _[Minor]_ In the initial motivating example, it is not clear what information of the "users" if any is being incorporated. Can this be clarified?
- **(Q3)** It is not clear what kind of "retrive, merge" operations would be performed for test examples/rows. Are these operations happening with training and test data combined? The handling of test data should be clearly and explicitly presented in my opinion.
  - If the test data augmentation is happening along with training data, how is data leakage handled in this setup?
  - If the test data augmentation happens separately for each test row, how is the consistency of features selected handled for the test rows, and how are the computational costs of retrieval/merge at test time incorporated in the runtime/performance tradeoff results?
- **(Q4)** _[MINOR]_ It is not clear how the number of failed configurations (across various sizes of the DL and Base Tables) are considered in the averages utilized in Figure 2.

**Strengths And Weaknesses:**

### Strengths
- **(S1)** Interesting and important practical problem, and the extensive evaluation with the accompanying benchmark is extremely valuable in identifying the important areas of future research.
- **(S2)** The paper covers appropriate amount of background information to help the reader clearly understand the different aspect of this problem setup, which is useful in understanding the insights presented in this paper.


### Weaknesses
- **(W1)** The present Pareto plots in Figure 2 are somewhat misleading in that it can lead to counterintuitive insights -- for example, in the "Retrival Method" column, it appears that "Exact" retrieval can some times be faster than the approximate "Minhash". This is of course not true, and an artifact that we are averaging over base tables and DLs (I think) and that the downstream classifier training time is included here. So exact search with say Catboost is overall faster than MinHash with ResNet. While this is true, it does not provide a clear ablation of the different components.
- **(W2)** Viewing Figure 2 (and table 3), it seems like "Exact" retrieval and "Full Join" selector are Pareto optimal in terms of resources-performances tradeoff. This seems somewhat disappointing. Any discussion in this observation will be useful otherwise the question is "why bother trying anything else if the exhaustive process is optimal on average". Is this highlighting an issue with the benchmark or the considered methods for the different steps, or the way the results are aggregated and presented? It seems unlikely that with all the recent methods developed in this area, the conclusion that "Exact Retrieval + Full Join" is Pareto-optimal needs better clarification.
- **(W3)** While discussion on page 10 claims that Minhash and hybrid-Minhash brings substantial improvements, Figure 2 (left) does not necessarily show them as Pareto optimal methods (unless I am missing something); this conclusion does not seem clear from the figures. In that case, there seems to be a bit of a discrepancy between the actual results and the discussion which needs further elaboration in the text.

---

> ### Author Response · Authors · 2025-02-24
>
> We thank the reviewer for the thorough feedback and suggestions.
>
> **W1: Pareto plots and run time of Exact matching vs MinHash.** The difficulty in reporting times is that certain operations are run more often than others, and we agree that we did not communicate well. Figure 2, 6, 8, 12, 13, 14, and 15 all report the times of the steps run in a cross-validation loop which dominates the costs (joint selection and predictor), not accounting for querying the data lake (retrieval), which is done once for a given target table. These times are reported in table 10. To report in the main figures timings that account for both steps, we propose to redo these figures with the times of retrieval + 10-fold cross-validation, corresponding to a typical data science run. Giving a fair accounting for retrieval time, such figures will show the speed benefit of MinHash. Note that merging the MinHash candidate produces tables from which learning is harder, leading to longer cross-validation times.
> The relative time of various retrieval methods depend on the usage patterns. Indeed, MinHash (and Starmie) come with an indexing step, costly but that can be amortized over different downstream analysis (table 10). We will modify the manuscript to be more explicit about what usage pattern a given figure reports and how different steps are called more or less frequently depending on usage patterns. We will also modify the paragraph on the computation trade-off to be more clear about the fact that index building can be considered as an offline cost, and can be amortized across multiple downstream tables.
>
> **W2.** While there has been much research in table retrieval, it has focused on creating a clean joined table, and evaluated as such. Supervised learning is very good at extracting signal from noise, and likely calls for different tradeoffs. Our benchmark explores a spectrum of techniques that could be used as a basis for further developments. Indeed, we hypothesize that there has not been enough focus on assembling tables from data lake for supervised learning, partly because of the lack of benchmark tasks.
>
> Different usage patterns also lead to different tradeoffs. For example, if the user does not have a clear idea of what is the correct column to join on, they may need to test a large number of them, thus making MinHash (or more efficient retrieval methods) more desirable.
>
> In addition, the performance of the selection algorithm depends on the specific case, and no configuration is always the best. For example, simple parametric methods like Ridge are strongly penalized with Full Join because of the presence of a large number of spurious features. Another example is Highest containment doing worse than all other methods in the YADL 50k case, which features a much larger degree of redundancy and thus is less likely to find the most relevant version of a table. The need to present synthetic figures in the main manuscript (for fear of overloading the reader), does not make these more nuanced messages obvious. We will add them to a discussion point.
>
> **W3.** Hybrid Minhash brings substantial improvements with respect to base Minhash, but it also retains some of its downsides. As the Pareto figures include only the training section of the pipeline, both methods are overshadowed by Exact matching. The real benefit of Minhash lies in its querying efficiency, which comes across more clearly in figure 5 and table 10. As we update main figures to show time of retrieval + 10-fold cross-validation, the tradeoffs will be more apparent from the figures.

---

> > ### Author Response · Authors · 2025-02-25
> >
> > A further example of how the downstream performance is affected by the downstream task can be highlighted by measuring the performance as a function of multiple variables. To this end, figures 21 and 22 include all possible combinations for the 5 main variables. In the following table (reporting the median of the Prediction performance, computed over all the other pipeline choices, for a choice of predictor and selector), we focus on the Selector-ML model pair to highlight how Full Join is indeed not always the best solution.
> >
> > |       Selector       | Catboost | RealMLP | ResNet | RidgeCV |
> > |:--------------------:|:--------:|:-------:|:------:|:-------:|
> > |   Best single join   |   0.418  |  0.146  |  0.087 |  0.107  |
> > |       Full join      |   0.451  |  **0.231**  |  0.058 |  -0.002 |
> > |  Highest containment |   0.383  |  0.146  |  0.066 |  0.048  |
> > | Stepwise greedy join |   **0.454**  |  0.216  |  **0.095** |  **0.158**  |
> >
> > It is hard to represent all the potential combinations in a way that is not overwhelming the reader, and we are open to suggestions on how to represent this information in a clearer way.

---

> > > ### Comment · Reviewer_DvRn · 2025-03-03
> > > **Nice discussion regarding the comments**
> > >
> > > I thank the authors for the insightful discussion in their responses. It completely aligns with what I would expect, and it would be great to have the nuances of the comparisons (as brought up by the authors) be explicitly discussion in the main text. This is of course a non-trivial task given the page limits, but I encourage the authors to think of how to do it best, and how to handle tradeoff between presenting results or discussion of the results in the main paper.
> > >
> > > I also appreciate the results (and in some cases, pointers to existing results) that highlight the suboptimality of the most naive baseline (Exact Match + Full Join). Again, highlighting these situations more explicitly in the main paper with demonstrate the room for improvement for future algorithm development. For example, the above table and the updated Figure 2 highlight this point well. Figure 17 is also very informative, and (at least to me) gives a nice overview of the results that compare the choices for the different steps across various data sets with respect to predictive performance (and I think computation time).
> > >
> > > **Note:** I think that the caption of Figure 17 needs to be updated to match the figure as it seems like the figure no longer matches the caption.
> > >
> > > Figures 21 and 22 are extremely informative, and can be seen as the "full evaluation" data in terms of the predictive performance. As the authors point out, this is a lot of information, and a great addition to the appendix. However, I urge the authors to accompany this figure with any insights that they want the reader to gleam from these figures. Maybe it is just a reiteration of some previous discussion, but pointing the reader to the specific insight(s) will allow the reader to make a direct connection, and extract the real value from having this figure in the appendix.
> > >
> > > The new (or in some cases, updated) explanatory figures (such as 1, 9) are also very helpful, and clarifies my questions with regards to how things are handled during train/test time. It is great that the scikit learn pipeline fit-predict pipeline is utilized here as that would ensure proper performance evaluation.
> > >
> > > Overall, I appreciate the detailed updates from the authors. I do not have any further comments beyond the above.

---

> > > > ### Author Response · Authors · 2025-03-10
> > > >
> > > > We thank the reviewer for the discussion and for suggesting ways to improve the manuscript. In fact, re-evaluating how we looked at the Pareto plot led to clearer takeaways, and to a deeper understanding of the problem.
> > > >
> > > > The current revision is not the final version of the paper and we will incorporate the comments and fixes suggested here and by the other reviewers, so that the interested reader may make proper use of the information contained in the main body and the appendix.

---

> ### Author Response · Authors · 2025-02-24
>
> **Q1.** Indeed, to our knowledge all existing retrieval methods produce join candidates agnostic to the downstream tasks. We agree with the reviewer that this is likely an important limitation that calls for more research. Regarding fuzzy matching, Starmie is a step in this direction, as it uses a column similarity metric that combines Jaccard similarity and the cosine similarity between embeddings of the columns, yet we did not investigate fuzzy joins (which do come with sizeable computational costs). We will add these two points to the discussion.
>
> **Q2.** In the initial motivating example, information on the users is not needed. We will remove the words “by users” of the first sentence to avoid confusion.
>
> **Q3.** Regarding test-time operations, new joins are not explored, but the joins selected at train time are applied. They are applied on the test rows only, at test time, and not jointly with the train rows. Note that this is a data augmentation, but in the feature (column) direction, and not in the sample (rows) direction.
>
> The consistency of features selected is ensured because the joins applied are exactly those selected at train time. If a match is not found for a given join, it creates a database Null, i.e. a missing value (this is also the case at train time). Specifically, we use the scikit-learn fit-predict paradigm to avoid data leakage between the training and testing split. Cross-validation is used to generate training and testing splits (20% test). Then, selection methods and supervised learners are trained using only the training split. To test the prediction performance, the held out test split from the cross-validation is passed to the selector, which then joins the test rows and uses that as an input to the internal ML model, which then yields a prediction.
>
> In terms of corresponding test-time compute cost, these are present in the figures (e.g. the pareto plot figures), as they report the average time to run a cross-validation fold.

---

### Review · Reviewer_HxpX · 2025-02-14

**Summary Of Contributions:**

In this paper, the authors investigate how to automate augmenting a user’s base table with features from other tables in a large data lake and then train a predictive model on the augmented table. The paper argues that no work studies the whole pipeline (retrieve, merge, and predict). They introduce a new semi-synthetic data lake called YADL, built by recombining data from the YAGO knowledge base. This controlled dataset allows for a systematic comparison of different augmentation and learning strategies. The authors also explore using an actual data lake, Open Data US, to validate their findings beyond synthetic settings. The experiment measures predictive performance, computational time, and memory usage. Their experiment results show that simple retrieval criteria based on set containment (exact) can work nearly as well as more advanced approaches. In addition, the result shows that CatBoost tolerate the noisy features and missing data introduced by automated table joins better than neural networks or linear baselines.

**Audience:**

Yes

**Broader Impact Concerns:**

No ethical concern.

**Claims And Evidence:**

No

**Requested Changes:**

1. Add more information about YADL, especially the prediction tasks, such as the number of base tables (ML Tasks), training and testing rows, and, if possible, a ceiling performance (if there is a perfect retrieval and merging).
2. Add more information about preprocessing steps of predictors.
3. Revise the discussion to be less duplicate with the result section.

**Strengths And Weaknesses:**

### Strengths

1. The paper provides the source code for reproducibility.
2. The paper proposes a new formulation of the machine-learning pipeline of multiple tables by expanding beyond feature synthesis and selection. I found that the proposed framework of "retrieve, merge, and predict" is interesting and well-suited to the challenges of big data analysis.

### Weaknesses

1. YADL is one of the paper's main contributions, but the paper lacks information regarding the supervised learning tasks. Thus, it is not clear which pipeline steps are "most" important for prediction performance (i.e., the ML tasks could be inherently difficult even with perfect selection and joining, or the ML tasks could be trivially simple once good columns have been joined.)
1. While the setting is about "limited memory," I found that the paper does not make this evidence in the proposed YADL data. For example, the size of the fully joined table might give a reader a rough idea of the challenge and the need to retrieve, merge, and predict the pipeline. In addition, the authors should acknowledge and discuss the work along the lines of big data architecture [1]. This direction of work might be similar to what authors try to achieve.
1. The conclusion regarding CatBoost is somewhat vague. The paper attributes the performance of CatBoost to the noise (null) and the cardinality of categorical features, but it has not been shown explicitly. In addition, preprocessing is an essential step for most predictive models, but the paper lacks this detail.
1. The main text of the paper could be more concise. This allowed more detail in the appendix to be in the main text.

### References

[1] M. Rong, D. Gong and X. Gao, "Feature Selection and Its Use in Big Data: Challenges, Methods, and Trends," in IEEE Access, vol. 7, pp. 19709-19725, 2019, doi: 10.1109/ACCESS.2019.2894366.

---

> ### Author Response · Authors · 2025-02-24
>
> We thank the reviewer for the thorough reading and feedback, and reply below to the main points.
>
> **Detailing the supervised learning tasks.** We will add more detail on the training procedure to the appendix. For all 6 base tables, we set aside 20% of the original samples as test split. Then, estimators that require an additional validation step (best single join, stepwise greedy join) perform an additional internal split where 20% of the training samples are kept as a validation set for evaluating the internal models. For these latter methods, in the final training step the best candidate (or set of candidates) is joined on the entire training set and the model is trained on the full set. In addition, all the pipeline steps seem important as visible from the ablation study (table 3): varying the reference method along each step leads to a marked variation in prediction, in order of importance: predictor (-26.0%), retrieval (-17.5%), aggregation (2.5%), and join selection (2.1%). The notion of perfect table is unclear: the exact answer to a prediction task may or may not be in a given data lake, finding it is the retrieval challenge. As a proxy of a good table, we can use the original downstream tables, with additional columns assembled by data scientists from other sources that we do not have access to.  We will add a table comparing the results obtained with the reference tables to those obtained by the automated retrieval methods.
>
> **Limited memory setting.** We will run additional experiments where we increase the number of possible candidates and provide a metric to estimate the memory footprint of joining all candidates. We also highlight that a larger memory footprint is only one of the effects of increasing the budget: a larger budget would also lead to much longer preparation/training time, and might also lead to the addition of poor candidates. We thank the reviewer for the reference and we consider it for our study of prior work.
>
> **Additional detail on Catboost.** We include new figures to the appendix (Figures 21 and 22) to highlight the difference in performance and interplay between different configurations. While Catboost reliably outperforms the parametric methods (Ridge, ResNet, RealMLP), this difference is more noticeable in those configurations that include more spurious features, such as the more redundant data lakes and the selector Full Join.
>
> **Additional detail on preprocessing.** For the parametric methods, all tables are preprocessed in the same way. Numerical columns are scaled, categorical columns with fewer than 30 unique values are one-hot encoded, categorical columns with more than 30 unique values are first one-hot encoded, and then PCA is applied to reduce the number of features to 30. Finally, missing values in all columns are imputed using the default scikit-learn SimpleImputer. CatBoost does not require any of these steps and can handle the input tables as-is. We will add a detailed section in the appendix on the preprocessing steps. We added a new diagram detailing the pre-processing step to the appendix (figure 10).

---

> > ### Comment · Reviewer_HxpX · 2025-03-01
> > **Clarification on Detailing the supervised learning tasks**
> >
> > **Detailing the supervised learning tasks.**
> >
> > I would like to clarify my concerns regarding this point. First, there is a need for evidence that the base tables of YADL can benefit from a good augmenting process (retrieving and join selection). The paper has demonstrated this indirectly by comparing it with a base configuration (exact retrieval and best-single join).
> >
> > Second, there is a need for evidence that the dataset is still challenging in different areas of the proposed framework. Table 3 shows that all of the steps in the pipeline can improve the prediction performance using the methods covered in the paper. We still lack evidence that there is room for improvement on retrieval or joining techniques (chaining joins as mentioned in Conclusion). Thus, the *perfect table* was recommended. However, I agree that
> >
> > >The notion of perfect table is unclear: the exact answer to a prediction task may or may not be in a given data lake, finding it is the retrieval challenge.
> >
> > A reference of an approximately good augmenting table based on the columns available in the data lake would be insightful.

---

> > > ### Author Response · Authors · 2025-03-01
> > >
> > > We thank the reviewer for the clarifications. We agree with the reviewer that the differences between retrieval methods provide evidence that the base tables benefit from augmenting from YADL. Regarding whether there is still room for improvement on the retrieval or joining, we agree that we had not established that the benchmark was not saturated, i.e. that it is impossible to find a better retrieval that improves downstream prediction. However, the questions and comments from the reviewers have led us to add an experiment varying the number tables retrieved. The results (figure 26 in the updated manuscript), show that retrieving more tables leads to improving downstream prediction, with a sizable performance increase compared to the original results with k=30. We will move this figure in the main manuscript and modify the discussion to state that YADL gives a benchmark to improve retrieval algorithms, striving to achieve best downstream performance with a limited number of candidates.
> > >
> > > We do not know how to build manually a silver standard. Engineering features from the data lake, or the original knowledge graph, YAGO, is difficult for a human being and requires much knowledge of the data and the application. One option to build such a silver standard would be to design an algorithm that semi-systematically tries combinations of the tables retrieved at k=100, probably with heuristics such as pruning. We will add this as further work.

---

> > > > ### Comment · Reviewer_HxpX · 2025-03-02
> > > > **Re: Clarification on Detailing the supervised learning tasks**
> > > >
> > > > Thank you for responding to my clarification. Figure 26 is insightful, and your suggested discussion will improve the manuscript.
> > > >
> > > > The revision has addressed the other requested changes well.

---

> > > > > ### Author Response · Authors · 2025-03-02
> > > > >
> > > > > We thank the reviewer for the fruitful exchange. The reviewer’s question did indeed lead us to thoughts and evidence that we hope will markedly improve the manuscript.

---

### Review · Reviewer_H1Bb · 2025-02-18

**Summary Of Contributions:**

The paper presents a comprehensive pipeline for automating feature augmentation in data lakes by decomposing the process into retrieval, join selection, aggregation, and prediction stages. In this work, the authors propose and evaluate multiple methods for each stage: retrieval methods based on Jaccard containment (including Exact Matching, MinHash, Hybrid MinHash, and the more computationally intensive Starmie), various join selectors (such as Highest Containment, Best Single Join, Stepwise Greedy Join, and Full Join), and aggregation techniques ranging from simple “any” and mean-based methods to the more elaborate DFS (Deep Feature Synthesis). The authors conduct an extensive experimental study across several datasets—including several variants of a semi-synthetic data lake (YADL) and Open Data US—to assess prediction performance alongside run-time and memory usage trade-offs. Their results indicate that while simpler, metric-based methods like Exact Matching coupled with straightforward join selection tend to achieve competitive predictive gains with lower computational cost, more complex methods (for instance, Starmie for retrieval or DFS for aggregation) incur significant overhead without commensurate improvements in downstream performance. Additionally, the experiments underscore that tree-based predictors (e.g., CatBoost) are particularly robust in handling the noise and high missingness introduced by automated table joins, compared to their neural network counterparts.

**Audience:**

Yes

**Broader Impact Concerns:**

None.

**Claims And Evidence:**

Yes

**Requested Changes:**

Please address the weaknesses.

**Strengths And Weaknesses:**

Strengths: The paper’s primary strength lies in its end-to-end treatment of the data augmentation problem in data lakes, an area that is both challenging and underexplored. By rigorously dissecting the pipeline into retrieval, join selection, aggregation, and prediction, the study offers clear insights into the interplay between different stages and the resulting performance trade-offs. The development of YADL—a controlled, semi-synthetic data lake—adds a significant experimental asset that allows for systematic exploration of factors such as table redundancy, join quality, and missing data. Moreover, the use of quantitative performance metrics, Pareto frontier analyses, and critical difference plots not only strengthens the experimental evaluation but also provides a transparent view of the resource versus performance trade-offs. The findings that simple retrieval techniques based on exact containment can outperform more elaborate methods in many scenarios, and that tree-based models handle the noise inherent in automated joins better than neural approaches, are both insightful and practically relevant.

Weaknesses: Despite the paper’s thorough experimental analysis, there are several technical limitations that warrant further discussion. First, the study confines itself to single-step join operations, thereby not addressing the complexities associated with multi-hop or chained joins that are common in heterogeneous real-world data lakes; this limitation reduces the generalizability of the findings to more complex relational structures. Second, while Jaccard containment is central to the retrieval methods, its reliance as a universal metric is problematic—especially in cases where columns have low cardinality or when semantic variations exist between join keys—yet the paper provides only a cursory treatment of alternative similarity measures or fuzzy matching techniques that might better capture such nuances. Third, the high computational cost associated with DFS-based aggregation is acknowledged, but the paper does not explore algorithmic optimizations or alternative methods that could alleviate this overhead, leaving a gap in the practical deployment of such approaches. Fourth, the experiments are conducted with a fixed candidate budget (e.g., selecting the top-30 candidates), but there is limited analysis on how varying this parameter would affect both scalability and predictive performance, potentially masking sensitivity issues in candidate selection. Finally, while the authors attempted to compare against several baseline methods, several state-of-the-art alternatives (such as Lazo, ALITE, and others) could not be fully integrated due to implementation challenges, which may constrain the comprehensiveness of the comparative evaluation and limit insights into the broader state of the art.

---

> ### Author Response · Authors · 2025-02-24
>
> We thank the reviewer for the thorough feedback, and reply below to the main points.
>
> An important context for our replies is the computational and operational cost of running the benchmarks, which total almost 200k CPU and GPU hours, despite not exploring the most sophisticated methods.
>
> **Multi-step joins.** We agree with the reviewer that considering only single-step and single-key joins is a limitation of our work. However, to the best of our knowledge, this work is the first to consider the problem of joining pipelines in the context of machine learning, and thus moves the literature forward. Furthermore, searching for combinations of keys or sequences of joins would increase the cost exponentially, quickly becoming intractable. This limitation is also shared by prior work in retrieval (Lazo [1], Auctus [2], Josie [3]).
>
> **Alternative similarity measures and fuzzy joining.** We agree that Jaccard containment is not a perfect metric and likely has some major limitations. This is a reason why we chose Starmie as an method to investigate: to include a method that would not rely exclusively on Jaccard containment and would instead be able to model similarity through some other metric (in Starmie’s case, the metric captures semantic links, using cosine similarity between embeddings). Yet, broadly speaking a similarity metric of some sort is needed to sort candidate joins. Given that the downstream join operation requires equality between the keys, useful metrics are likely to share some properties of the Jaccard containment.
>
> **Algorithmic optimizations of DFS.** Developing an optimized version of DFS was beyond the scope of our work, as it is a benchmark, thus it considered only prior work. Our experimental results show that DFS brings a noticeable improvement to the downstream performance, at a large cost in execution time. As a result, an optimized version of the DFS code with a better execution efficiency would likely bring it closer to the execution cost of the simpler methods, while retaining the better prediction performance. We are aware of a recent work ([4]) which claims to improve over the original DFS code, adding to the original DFS code more aggregation functions and time-wise cross-validation. While optimizations such as query optimization with a database propositionalization engine can indeed decrease the computational cost, and improve the Pareto optimality of the method, they are most relevant for “deep” explorations (multi-step joins), which we did not benchmark here. Hence, we do not expect completely different conclusions even with more advanced implementations of the ideas in DFS.
>
> **Testing the impact of the size of the budget.** We thank the reviewer for the excellent suggestion. We will add an experiment running the reference configuration using different values of K to track how the performance changes as more candidates are considered.
>
> [1] Fernandez, Raul Castro, et al. "Lazo: A cardinality-based method for coupled estimation of jaccard similarity and containment." 2019 IEEE 35th International Conference on Data Engineering (ICDE). IEEE, 2019.
> [2] Castelo, Sonia, et al. "Auctus: A dataset search engine for data augmentation." arXiv preprint arXiv:2102.05716 (2021).
> [3] Zhu, Erkang, et al. "Josie: Overlap set similarity search for finding joinable tables in data lakes." Proceedings of the 2019 International Conference on Management of Data. 2019.
> [4] Wang, Minjie, et al. "4dbinfer: A 4d benchmarking toolbox for graph-centric predictive modeling on relational dbs." arXiv preprint arXiv:2404.18209 (2024).

---

### Author Response · Authors · 2025-02-24

We thank the reviewers for their reviews and their insightful comments.

We note that all reviewers engaged with precise questions and comments, asking for clarifications of our benchmarking choices or our results. We reply to these in the reply to each reviewer.

In addition, in the light of the reviewers comments and questions, and to bring material to our reply, we have added to the appendix the following:
- A diagram that explains the cross-validation setup and how training and testing samples are handled (Appendix section C.2, Figure 9).
- A diagram that explains the pre-processing applied to the tables that are passed to the ML models (Appendix section C.2, Figure 10).
- New figures that show the interplay between variables (appendix Figures 21 and 22).

---

> ### Author Response · Authors · 2025-03-01
>
> We further revise the manuscript by adding to the appendix:
> - New figures that show how prediction performance, peak RAM at fit time, and fold runtime are affected by the number of retrieved candidates (figures 23, 24, 25 respectively).
> - A new Pareto plot to show the prediction performance-run time trade-off as a function of top-k (figure 26).
>
> We also revise the Pareto plots in figures 2 and 14: rather than reporting the runtime of each fold, we instead model a different scenario in which the retrieval (query) time is added to the runtime of the full cross-validation operation. Furthermore, we modify the Pareto plot for the RAM to show the Peak RAM for each method.
> In the same vein, we modify table 3 to measure the difference from the reference time after adding the query step.
> By doing so, the impact of retrieving candidates is accounted for, so that faster retrieval methods are represented more fairly.

---

### Decision · Action_Editor_HVHM · 2025-04-04

**Recommendation:** Accept as is

**Comment:**

The reviewers found that the problem studied is "challenging and underexplored" (Reviewer H1Bb), "interesting and important" (Reviewer DvRn), the proposed framework is "interesting and well-suited to the challenges of big data analysis" (Reviewer HxpX). They also found that the new data lake offers "a significant experimental asset that allows for systematic exploration of factors" (Reviewer H1Bb) and is "extremely valuable in identifying the important areas of future research." (Reviewer DvRn). Reviewer H1Bb also noted that the findings are "insightful and practically relevant".

During the rebuttal, the authors and reviewers had fruitful exchanges around discussion of limitations and future work, clarified additional details about the tasks and settings, discussed nuances of different comparisons, the suboptimality of naive baselines and clarifying where there is room for improvement. I encourage the authors to reflect these discussions in the main paper and include an explicit section on weaknesses of the existing investigation to guide future work.

**Audience:**

Yes, all reviewers found that the paper studies an important and practically relevant problem and will be of interest to the TMLR community.

**Claims And Evidence:**

This paper investigates approaches for augmenting a base table with features from other relevant tables in a data lake for the purpose of obtaining training data for machine learning models. They also introduce a new data lake for benchmarking retrieval and augmentation methods. They then implement a full pipeline of "retrieve, merge, predict" combining different strategies for each step and present an empirical investigation in terms of prediction performance, execution time and memory usage.

The reviewers found that the claims are supported by a series of experiments varying retrieving methods, merge methods, and machine learning algorithms (Reviewer HxpX). Reviewer H1Bb said: "By rigorously dissecting the pipeline into retrieval, join selection, aggregation, and prediction, the study offers clear insights into the interplay between different stages and the resulting performance trade-offs." and "the use of quantitative performance metrics, Pareto frontier analyses, and critical difference plots not only strengthens the experimental evaluation but also provides a transparent view of the resource versus performance trade-offs."

Reviewer HxpX pointed out incomplete evidence that the proposed new data lake is still challenging in terms of all aspects of the pipeline of interest. The authors addressed this to the reviewer's satisfaction by adding a new experiment that shows room for improvement (improvement to downstream prediction by retrieving more tables).